# Reference-Guided Machine Unlearning

## Abstract

Machine unlearning aims to remove the influence of specific training data from a model while preserving its general utility. In vision, many approximate unlearning methods pursue this goal through degradation-based heuristics, such as loss maximization or random labeling. Yet making a model worse on forget samples is not the same as making it behave as if those examples had never been seen: these signals can be poorly conditioned, destabilize optimization, and harm generalization. We argue that approximate unlearning should instead prioritize distributional indistinguishability, aligning the model's predictive behavior on forget data with that on truly unseen data. Motivated by this principle, we propose Reference-Guided Unlearning (REGUN), a vision unlearning framework that uses disjoint held-out data to construct a principled, class-conditioned reference distribution for distillation. Rather than explicitly degrading predictions on forget examples, REGUN guides them toward non-member behavior through held-out supervision. Across multiple architectures, natural image datasets, and forget fractions, REGUN consistently improves the forgetting–utility trade-off over standard approximate baselines while closely matching retrain-like membership inference behavior. As one instantiation of this principle, the results suggest that simple objectives designed around indistinguishability can provide a stronger and more stable alternative to complex degradation-based unlearning procedures.

## 1 Introduction

Machine unlearning is the principled process of updating a trained model to remove the influence of specified forget examples. It has become an operational requirement for deployed AI systems, driven by privacy regulations such as the European Union's GDPR "right to be forgotten" (Mantelero, 2013) or the California Consumer Privacy Act (California State Legislature, 2018), as well as the operational need to adapt models post-deployment (Nguyen et al., 2025). While retraining the model from scratch without the forget examples provides the most faithful solution, it is often computationally prohibitive at scale. As a result, most unlearning methods rely on approximate updates (e.g., short fine-tuning or deletions) to emulate this retraining by reducing reliance on the forget data, but doing so without sacrificing retained utility remains challenging (Triantafillou et al., 2024).

A common strategy in approximate unlearning is to degrade model performance on forget examples, for instance, through loss maximization or random and pseudo-label supervision (Li et al., 2025). However, such objectives can be poorly conditioned: they may induce large or misdirected gradients that change decision boundaries beyond the intended region and harm generalization (Mavrothalassitis et al., 2025). To mitigate this damage, many methods introduce additional constraints, e.g., remaining close to the original model (Kurmanji et al., 2023), applying repair mechanisms (Tarun et al., 2024), or using constrained parameter editing (Fan et al., 2024; Foster et al., 2024). This reveals a central mismatch: degradation objectives optimize for forgetting-like symptoms, but do not necessarily mimic unfamiliarity, i.e., the behavior expected from a model that had never seen the forget data.

**Forgetting should mean indistinguishability, not degradation.** Rather than merely making the model "more wrong", we argue that unlearning should align its behavior on forget data with that of truly unseen examples. While this indistinguishability idea has been considered (Thudi et al., 2022b), a practical reference

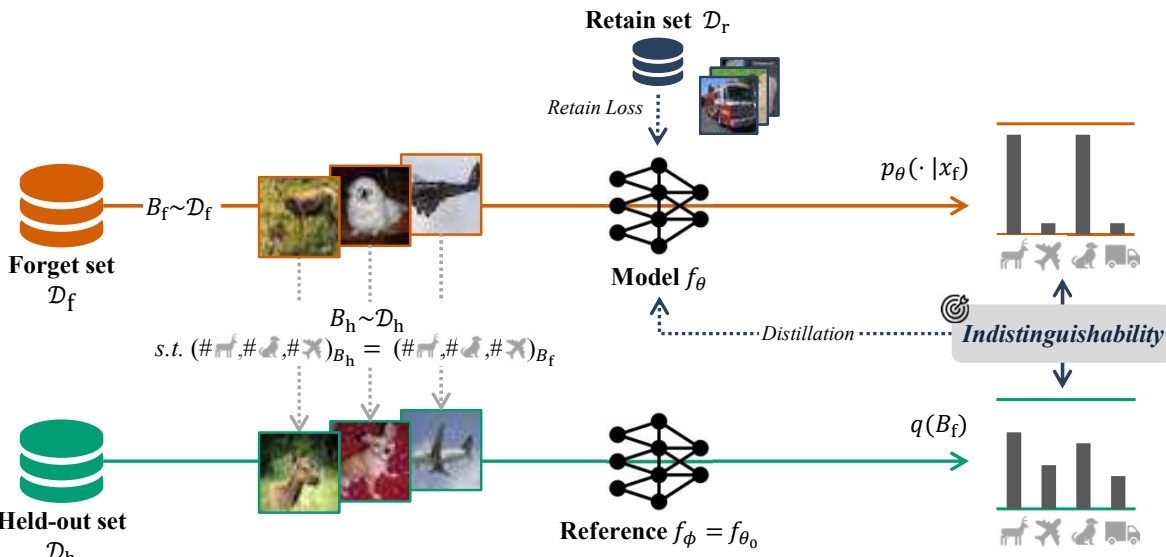

Figure 1: **Unlearning optimization in ReGUn.** In REGUN, unlearning follows a reference-guided distillation objective based on *indistinguishability*. Given a forget minibatch $B_f \sim \mathcal{D}_f$, REGUN samples a held-out minibatch $B_h \sim \mathcal{D}_h$ such that the class histogram of $B_h$ matches that of $B_f$. The samples in $B_h$ are passed through a frozen reference model to construct a reference distribution $q$, which represents the desired "unseen" behavior for the forget examples. The current model $f_\theta$ is then updated by distilling its predictions on $B_f$ toward this reference distribution, encouraging forget samples to become indistinguishable from non-member data. In parallel, a standard cross-entropy loss on retain samples is used to preserve model utility.

for "unseen behavior" remains missing. We bridge this gap by proposing the paradigm of REference-Guided UNlearning (REGUN)[1] with held-out supervision. REGUN uses a disjoint held-out dataset as a stable proxy for "unseen behavior", aligning the model's outputs on forget examples with this reference distribution, as illustrated in Figure 1. Crucially, this target is neither a uniform distribution, nor a random label, nor the original prediction of the model on the forget example itself. Instead, it is a model-induced distribution obtained by evaluating a frozen reference model on disjoint held-out samples with the same class composition as the forget batch. Thus, the forget examples are not pushed toward artificial uncertainty or corrupted supervision, but toward the average predictive behavior assigned to unseen samples from the relevant classes.

Prior reference-based methods supervise unlearning by replacing forget-sample outputs with pseudo-probabilities (e.g., uniform or global distributions) (Zhao et al., 2024), or by aligning them with generic third-party data (sometimes even unrelated noise), yielding a coarse, non-class-aware, and potentially noisy unlearning signal (Chen et al., 2025). In contrast, REGUN uses a disjoint held-out subset as an explicit supervision source to construct the reference, enabling class-conditioned references rather than only marginal distribution matching. Finally, although a portion of the literature studies *class*-level removal (Zhang et al., 2025; Gandikota et al., 2023), we focus on approximate *instance*-level unlearning, which is more closely aligned with individual privacy and consent. This work targets the vision domain, building on the extensive body of literature (Ebrahimpour-Boroojeny et al., 2025; Hsu et al., 2025; Khalil et al., 2025; Fan et al., 2024; Lee et al., 2025) that focuses on unlearning for vision.

**Contributions.** Our main contributions to the field of machine unlearning include:

- We motivate that approximate unlearning should optimize for indistinguishability from unseen data, rather than relying on objectives that mainly degrade forget-set performance.
- We introduce REGUN, a reference-guided unlearning framework for vision that distills forget-set predictions toward class-conditioned distributions constructed from disjoint held-out data.
- We empirically validate REGUN across multiple image-classification benchmarks, architectures, and forget fractions, showing improved forgetting–utility trade-offs over the studied baselines.

---

[1] Code available at: `https://anonymous.4open.science/r/ReGUn-7E0B`

## 2 Related Work

**Machine Unlearning.** Machine unlearning aims to remove the influence of specific training samples from a trained model while preserving utility on retained data. Early work focused primarily on exact unlearning methods with formal guarantees, including retraining-based approaches and data partitioning strategies such as SISA (Cao & Yang, 2015; Bourtoule et al., 2021; Guo et al., 2020). While theoretically appealing, these approaches remain computationally expensive and difficult to scale to modern deep learning settings. As a result, recent research has increasingly focused on approximate unlearning methods (Nguyen et al., 2025; Laguna et al., 2026). A dominant family of approaches performs post-hoc parameter updates designed to degrade model performance on forget samples. These include catastrophic-forgetting-based methods (Goel et al., 2023), gradient ascent approaches such as NegGrad and NegGrad+ (Kurmanji et al., 2023), label perturbation strategies (Graves et al., 2021), sparsity-based updates (Jia et al., 2023), parameter dampening methods such as SSD (Foster et al., 2024), saliency-guided editing (Fan et al., 2024), adversarial forgetting approaches (Ebrahimpour-Boroojeny et al., 2025), and contrastive formulations (Khalil et al., 2025). Distillation-based approaches, including Bad Teacher (Chundawat et al., 2023) and SCRUB (Kurmanji et al., 2023), attempt to repair utility during forgetting by transferring knowledge from auxiliary teacher models.

Despite their differences, these methods largely share a common paradigm: forgetting is typically encouraged through objectives that reduce, perturb, or disrupt model behavior on forget examples. However, degrading predictions does not necessarily imply that the resulting model behaves like one that never observed the forgotten data. In practice, these optimization signals can be poorly conditioned and may induce unnecessary changes to retained representations. Beyond standard instance-level unlearning, recent work has also studied related settings such as graph and recommendation unlearning (Chen et al., 2022b;a), incremental unlearning in pre-trained vision models (Feng et al., 2025; Kawamura et al., 2025; Wang et al., 2025), and concept unlearning in generative or foundation models (Gandikota et al., 2023; Gao et al., 2025; Biswas et al., 2025; Zhang et al., 2025; Dong et al., 2025). While these directions broaden the scope of machine unlearning, our work focuses on approximate instance-level unlearning for discriminative vision models, where the central challenge is balancing forgetting efficacy with retained utility.

**Indistinguishability in Unlearning.** A central objective of unlearning is to approximate the behavior of a model retrained without the forget set. Consequently, successful unlearning is often implicitly measured through indistinguishability: forget examples should become statistically indistinguishable from unseen data (Thudi et al., 2022a), rather than from retained training samples. In practice, this principle is commonly evaluated through membership inference attacks (MIAs) (Shokri et al., 2017; Zarifzadeh et al., 2024), where successful forgetting corresponds to the inability to distinguish forget samples from non-members.

However, most approximate methods are not optimized for indistinguishability directly, creating a mismatch between the optimization objective and the privacy behavior ultimately being evaluated. REGUN aims to close this gap by directly optimizing toward unseen behavior using held-out references. A model may become less accurate on forget examples while still retaining detectable membership signals or residual information about the removed data. Recent work has increasingly highlighted this tension between forgetting strength and utility preservation (Mavrothalassitis et al., 2025; Triantafillou et al., 2024). Our work builds on this perspective and turns indistinguishability from an evaluation criterion into an optimization signal by distilling forget-set predictions toward held-out non-member behavior.

**Reference Guidance.** Several prior methods use auxiliary output distributions to guide unlearning. Pseudo-probability methods replace forget-set predictions with artificial targets, such as uniform probability distributions (Zhao et al., 2024). Others align forget-set outputs with generic third-party data, sometimes even unrelated noise, producing a marginal, non-class-aware forgetting signal (Chen et al., 2025). Distillation-based methods use teacher-student objectives primarily to stabilize the forgetting process and preserve retain-set utility. In particular, Bad Teacher (Chundawat et al., 2023) distills incorrect behavior on forget samples through an incompetent teacher, whereas SCRUB (Kurmanji et al., 2023) combines forgetting objectives with distillation toward the original model on retained data. In contrast, REGUN uses same-dataset held-out samples to construct an explicit class-conditioned non-member target. This makes the target more specific: forget examples should behave like unseen samples from the same class composition, rather than being pushed toward uniform, generic third-party, noise-induced, or teacher-modified examples.

# 3 ReGUn: Reference-Guided Machine Unlearning

## 3.1 Problem Formulation

We consider supervised $K$-class classification with input space $\mathcal{X}$ and labels $\mathcal{Y} = \{1, \ldots, K\}$, and write $\mathcal{D} = \{(x_i, y_i)\}_{i=1}^n \subseteq \mathcal{X} \times \mathcal{Y}$ for a labeled dataset. With $\Delta^K$ being the probability simplex over $K$ classes, let $f_\theta : \mathcal{X} \to \Delta^K$ be a probabilistic classifier with parameters $\theta$ and predictive distribution $p_\theta(\cdot \mid x) = f_\theta(x)$. Machine unlearning starts with a model trained on $\mathcal{D}_{\mathrm{train}}$ and a request to remove the influence of a designated forget set $\mathcal{D}_{\mathrm{f}} \subset \mathcal{D}_{\mathrm{train}}$. We write $\mathcal{D}_{\mathrm{train}} = \mathcal{D}_{\mathrm{r}} \cup \mathcal{D}_{\mathrm{f}}$ with $\mathcal{D}_{\mathrm{r}} \cap \mathcal{D}_{\mathrm{f}} = \emptyset$, where $\mathcal{D}_{\mathrm{r}}$ is the retain set. Let $\theta_0$ be obtained by training on $\mathcal{D}_{\mathrm{train}}$, before any unlearning request is applied. The goal of machine unlearning is to produce parameters $\theta_{\mathrm{u}}$ such that $f_{\theta_{\mathrm{u}}}$ behaves like the retraining baseline that would result if $\mathcal{D}_{\mathrm{f}}$ had never been seen. Concretely, letting $\theta_{\mathrm{r}}$ denote parameters obtained by training on $\mathcal{D}_{\mathrm{r}}$ only from scratch, we aim for $f_{\theta_{\mathrm{u}}} \approx f_{\theta_{\mathrm{r}}}$, while avoiding full retraining. This approximation is empirical rather than certified and does not provide a formal guarantee of equivalence to retraining. Finally, we assume access to a disjoint held-out labeled set $\mathcal{D}_{\mathrm{h}} = \{(x_j, y_j)\}_{j=1}^{n_{\mathrm{h}}}$ with $\mathcal{D}_{\mathrm{h}} \cap \mathcal{D}_{\mathrm{train}} = \emptyset$, which is used only to construct reference distributions during unlearning.

## 3.2 Unlearning from Reference Distributions

Rather than corrupting forget-set predictions, we aim to align them with the behavior corresponding to inputs never seen by the model. Ideally, this corresponds to matching the unavailable retrained model $f_{\theta_{\mathrm{r}}}$ on the forget set, i.e., minimizing $\mathbb{E}_{(x,y) \in \mathcal{D}_{\mathrm{f}}}[\mathrm{KL}(p_{\theta_{\mathrm{r}}}(\cdot \mid x) \| p_\theta(\cdot \mid x))]$. Since $\theta_{\mathrm{r}}$ is precisely the retrain-from-scratch solution that approximate unlearning seeks to avoid, we instead use the held-out set $\mathcal{D}_{\mathrm{h}}$ to operationalize this "unseen behavior". We thus treat unlearning as distilling forget-set predictions to match this reference distribution, as detailed in Algorithm 1 and illustrated in Figure 1.

---

**Algorithm 1** REGUN: Reference-Guided Unlearning

1: **Input:** initial model $f_{\theta_0}$, retain $\mathcal{D}_{\mathrm{r}}$, forget $\mathcal{D}_{\mathrm{f}}$, held-out $\mathcal{D}_{\mathrm{h}}$, weights $\lambda_{\mathrm{f}}, \lambda_{\mathrm{r}}$, steps $T$, operator REFDIST$_{\mathrm{cls}}$
2: Initialize $\theta \leftarrow \theta_0$
3: Set $f_\phi \leftarrow f_{\theta_0}$                         ▷ default; alternative $f_\phi$s ablated in Sec. 5.
4: **for** $t = 1, \ldots, T$ **do**
5:      Sample minibatches $B_{\mathrm{f}} \sim \mathcal{D}_{\mathrm{f}}$, $B_{\mathrm{r}} \sim \mathcal{D}_{\mathrm{r}}$
6:      $q(B_{\mathrm{f}}) \leftarrow$ REFDIST$_{\mathrm{cls}}(B_{\mathrm{f}}; \mathcal{D}_{\mathrm{h}}, f_\phi)$           ▷ default; alternative REFDISTs ablated in Sec. 5.
7:      $\mathcal{L}_{\mathrm{f}} \leftarrow \frac{1}{|B_{\mathrm{f}}|} \sum_{(x,\cdot) \in B_{\mathrm{f}}} \mathrm{KL}(q(B_{\mathrm{f}}) \| p_\theta(\cdot \mid x))$
8:      $\mathcal{L}_{\mathrm{r}} \leftarrow \frac{1}{|B_{\mathrm{r}}|} \sum_{(x,y) \in B_{\mathrm{r}}} \mathrm{CE}(p_\theta(\cdot \mid x), y)$
9:      $\mathcal{L} \leftarrow \lambda_{\mathrm{f}} \mathcal{L}_{\mathrm{f}} + \lambda_{\mathrm{r}} \mathcal{L}_{\mathrm{r}}$
10:     Update $\theta$ with one optimizer step on $\mathcal{L}$
11: **end for**
12: $\theta_{\mathrm{u}} \leftarrow \theta$
13: **Output:** unlearned model $f_{\theta_{\mathrm{u}}}$

---

**Reference Distribution.** At each iteration in the unlearning phase, we randomly sample a forget mini-batch $B_{\mathrm{f}} \sim \mathcal{D}_{\mathrm{f}}$ and compute a batch-level soft target as detailed in Algorithm 2 via REFDIST$_{\mathrm{cls}}$:

$$q(B_{\mathrm{f}}) := \mathrm{REFDIST}_{\mathrm{cls}}(B_{\mathrm{f}}; \mathcal{D}_{\mathrm{h}}, f_\phi) \in \Delta^K, \tag{1}$$

where $f_\phi$ is a reference model. Ideally, $f_\phi = f_{\theta_{\mathrm{r}}}$, since this oracle model best represents the desired post-unlearning behavior. However, $f_{\theta_{\mathrm{r}}}$ is unavailable in approximate unlearning. We therefore set $f_\phi = f_{\theta_0}$, the initial model state, as a proxy to avoid extra training and prevent reference drift. Importantly, although the reference model $f_{\theta_0}$ still retains influence from $\mathcal{D}_{\mathrm{f}}$, REGUN never distills forget examples toward their own pre-unlearning predictions; the reference is computed only from disjoint held-out inputs. To construct the reference target $q(B_{\mathrm{f}})$, REFDIST$_{\mathrm{cls}}$ samples a held-out batch $B_{\mathrm{h}} \sim \mathcal{D}_{\mathrm{h}}$ with the same class histogram as $B_{\mathrm{f}}$, and aggregates its predictions under $f_\phi$ into a single distribution. This controls for label-prior differences and yields a batch-level estimate of unseen behavior without matching forget examples to individual

held-out counterparts. These defaults are not strict requirements for REGUN. In Section 5, we ablate both components of the reference construction: the reference model $f_\phi$ and the selector REFDIST used to sample $B_h$. Specifically, we compare the frozen teacher $f_{\theta_0}$ against online $f_\theta$, its Exponential Moving Average (EMA) $f_{\text{EMA}}$, and oracle retrained $f_{\theta_r}$ teachers, and compare class-matched sampling against uniform and feature-nearest-neighbor held-out selectors.

To aggregate the predictions from $B_h$, we evaluate the reference model $f_\phi$ on each held-out input and average the resulting predictive distributions:

$$q(B_f) \;=\; \frac{1}{|B_h|} \sum_{(x,\cdot) \in B_h} p_\phi(\cdot \mid x), \tag{2}$$

which corresponds to an empirical mixture of the reference model's outputs on the selected held-out inputs. The same $q(B_f)$ is used for all $x \in B_f$ and, notably, it depends on $B_f$ through batch statistics, i.e., the label histogram, rather than through instance-wise matching. This batch-level construction avoids introducing noise from potentially brittle pairwise matching between forget and held-out examples. Algorithm 2 provides the step-by-step formulation of the reference distribution. Although REGUN can be viewed as non-member distillation, its target is not a hand-designed entropy prior such as a uniform distribution, random label, or label-smoothed target (Müller et al., 2019; Chundawat et al., 2023). Instead, it is estimated from reference-model predictions on disjoint examples, yielding a class-conditioned non-member target that preserves model-specific uncertainty while avoiding distillation from the original forget-sample predictions.

---

**Algorithm 2** REFDIST$_{\text{cls}}$: Class-Conditioned Held-out Reference Distribution

1: **Input:** forget batch $B_f$, held-out set $\mathcal{D}_h$, reference model $f_\phi$
2: Let $c_k = \sum_{(\cdot,y) \in B_f} \mathbf{1}[y = k]$ for $k \in \{1, \dots, K\}$
3: Let $\mathcal{D}_h^{(k)} = \{(x,y) \in \mathcal{D}_h : y = k\}$ for $k \in \{1, \dots, K\}$
4: Sample $B_h$: draw $c_k$ examples uniformly from $\mathcal{D}_h^{(k)}$ for each class $k$ (with replacement if needed)
5: Aggregate reference predictions: $q \leftarrow \frac{1}{|B_h|} \sum_{(x,\cdot) \in B_h} p_\phi(\cdot \mid x)$
6: **Output:** $q \in \Delta^K$

---

**Unlearning Objective.** Given the reference distribution $q(B_f)$, unlearning is performed by optimizing two complementary terms. The forget term aligns the model predictions on forget inputs with the held-out reference distribution through a KL loss, encouraging forget examples to behave like non-member data. In parallel, a retain term anchors unlearning to preserve utility on retained data by applying standard cross-entropy on retain samples. For a retain minibatch $B_r \sim \mathcal{D}_r$, REGUN minimizes

$$\mathcal{L}(\theta; B_f, B_r) \;=\; \lambda_f \frac{1}{|B_f|} \sum_{(x,\cdot) \in B_f} \text{KL}(q(B_f) \,\|\, p_\theta(\cdot \mid x)) \;+\; \lambda_r \frac{1}{|B_r|} \sum_{(x,y) \in B_r} \text{CE}(p_\theta(\cdot \mid x), y), \tag{3}$$

where $\lambda_f, \lambda_r > 0$ trade off forgetting strength and retain utility, and $\text{CE}(p, y) = -\log p(y)$ is the standard cross-entropy for a hard label $y$. Since $q(B_f)$ is fixed with respect to $\theta$ within each update, minimizing the forward $\text{KL}(q(B_f)\|p_\theta(\cdot \mid x))$ is equivalent to minimizing the cross-entropy with the held-out soft target $q(B_f)$ up to an additive constant. Thus, the forget term can be interpreted as standard distillation toward a held-out teacher distribution.

## 4 Experimental Setup

**Datasets.** We evaluate REGUN on three standard image classification benchmarks (Li et al., 2025) to analyze the forgetting–utility trade-off: CIFAR-10, CIFAR-100 (Krizhevsky, 2009), and Tiny-ImageNet (Stanford CS231N, 2015). CIFAR-10 and CIFAR-100 each contain 60,000 $32 \times 32$ color images split into 50,000 training and 10,000 test images, with 10 and 100 classes, respectively. Tiny-ImageNet contains 100,000 training and 10,000 validation images, used for testing, at $64 \times 64$ resolution across 200 classes. For Transformer-based architectures, the images are upsampled to $224 \times 224$ resolution, whereas CNN baselines utilize the native dimensions. We outline additional implementation details and augmentation pipelines in Appendix A.1.

Following our unlearning protocol, we split the original training set $\mathcal{D}_{\mathrm{orig}}$ into disjoint subsets using a seeded random split. We first reserve a validation split $\mathcal{D}_{\mathrm{val}}$ of size $0.1|\mathcal{D}_{\mathrm{orig}}|$ for hyperparameter selection. From the remaining pool $\mathcal{D}_{\mathrm{orig}} \setminus \mathcal{D}_{\mathrm{val}}$, we sample the forget set $\mathcal{D}_{\mathrm{f}}$ uniformly at random with $|\mathcal{D}_{\mathrm{f}}|/|\mathcal{D}_{\mathrm{orig}} \setminus \mathcal{D}_{\mathrm{val}}| \in \{1\%, 10\%, 50\%\}$, and reserve a held-out set $\mathcal{D}_{\mathrm{h}}$ of size $0.1|\mathcal{D}_{\mathrm{orig}} \setminus \mathcal{D}_{\mathrm{val}}|$. The remaining samples form the retain set $\mathcal{D}_{\mathrm{r}}$. The base model is trained from scratch on $\mathcal{D}_{\mathrm{train}} = \mathcal{D}_{\mathrm{r}} \cup \mathcal{D}_{\mathrm{f}}$ and then unlearned. We assume access to labels for the retained, forgotten, and held-out splits during unlearning.

**Baselines.** We compare against a diverse set of approximate unlearning methods spanning gradient-ascent, fine-tuning, sparsification, learning-to-unlearn, parameter-dampening, saliency-based, and adversarial approaches. We include NEGGRAD and NEGGRAD+ (Kurmanji et al., 2023), which apply gradient ascent on forget samples, with NEGGRAD+ additionally using retain-set supervision. We evaluate FINETUNE, which continues training only on the retain set, and $\ell_1$-SPARSE (Jia et al., 2023), which augments retain-set fine-tuning with an $\ell_1$ sparsity penalty on the model parameters. We further include LUR (Patel & Qiu, 2025), which uses a learning-to-unlearn objective to mitigate retain–forget gradient conflicts. Notably, for the Transformer-based setting, we use an adapted implementation described in Appendix A.1. We also compare against SSD (Foster et al., 2024), an optimization-free Fisher-dampening method that suppresses parameters associated with forget samples. To cover localization-based unlearning, we include SALUN (Fan et al., 2024), which constructs gradient-based weight-saliency masks and restricts random-label unlearning updates to the salient parameters while preserving retain-set utility. Finally, we include AMUN (Ebrahimpour-Boroojeny et al., 2025) as an adversarial unlearning baseline that uses adversarial examples during unlearning to enforce forgetting while maintaining utility on retained data. Together, these baselines cover representative degradation-, localization-, and regularization-based approximate unlearning strategies. As reference, we also report results for $f_{\theta_{\mathrm{r}}}$ (RETRAIN) and $f_{\theta_0}$ (BASE).

**Models and Experimental Settings.** In the main text, we report two model–dataset settings, covering both standard CNN-based image classification and a higher-resolution Transformer-based benchmark: ResNet-18 (He et al., 2016) on CIFAR-10, and Swin-T (Liu et al., 2021) on Tiny-ImageNet (Stanford CS231N, 2015). ResNet-18 is a compact residual CNN with approximately 11M parameters, providing a standard convolutional backbone for low-resolution image classification. By contrast, Swin-T is a hierarchical vision Transformer with approximately 28M parameters based on shifted-window self-attention. Results on two additional settings with ResNet-18 on CIFAR-100 and ResNet-18 on Tiny-ImageNet are provided in Appendix B.2. All training details, optimizer settings, and hyperparameter searches for REGUN and the studied baselines are reported in Appendix A.1.

**Metrics.** We evaluate unlearning along its two core objectives: utility preservation and forgetting efficacy. For utility, we report retain accuracy (RA) on $\mathcal{D}_{\mathrm{r}}$ and test accuracy (TA) on the test set. For forgetting, we report forget accuracy (FA) on $\mathcal{D}_{\mathrm{f}}$ and membership inference performance. For membership inference, we use an offline likelihood-ratio score based on robust MIA (RMIA) (Zarifzadeh et al., 2024): For each candidate example $x$ with label $y$, we compute the log-ratio between the target model's true-label probability $p_\theta(y \mid x)$ and the average true-label probability assigned by four attack reference models trained on $\mathcal{D}_{\mathrm{r}}$ and therefore without the forget examples. We report the receiver operating characteristic area under the curve (ROC-AUC) of this attack score, denoted $\mathrm{RMIA}_{\mathrm{AUC}}$. Values close to 50% indicate chance-level discrimination between forget samples and non-members, matching the indistinguishability criterion underlying unlearning. To summarize performance across metrics, we follow existing work (Patel & Qiu, 2025; Fan et al., 2024) and report $\mathrm{Gap}_{\mathrm{Avg}}$. This metric averages the deviation from the retrain-from-scratch baseline (RETRAIN) across RA, TA, FA, and $\mathrm{RMIA}_{\mathrm{AUC}}$. Lower values indicate closer agreement with the retraining oracle. Further, to capture the utility-forgetting trade-off more directly, we treat TA and $\mathrm{RMIA}_{\mathrm{AUC}}$ as the primary representatives of utility and forgetting, respectively. We use these two metrics in the trade-off analysis and also report $\mathrm{Gap}_{\mathrm{Avg}}^{\mathrm{F\text{-}U}}$, defined as the average gap to RETRAIN over TA and $\mathrm{RMIA}_{\mathrm{AUC}}$. Finally, we include a standard loss-based MIA ($\mathrm{SMIA}_{\mathrm{AUC}}$) and its corresponding $\mathrm{Gap}_{\mathrm{Avg}}^{\mathrm{SMIA}}$, defined as the same aggregate gap but replacing RMIA with SMIA. This complements the RMIA-based evaluation with a simpler loss-threshold diagnostic, closer in spirit to the MIA-style scores commonly used in unlearning evaluations (Chundawat et al., 2023; Foster et al., 2024; Kurmanji et al., 2023) and grounded in standard loss-based membership inference (Carlini et al., 2022). All results reported in this manuscript are averaged over three seeds and reported as mean $\pm$ std, with all metrics expressed in %.

| | RA | FA | TA | $\text{RMIA}_{\text{AUC}}$ | $\text{Gap}_{\text{Avg}}$ | $\text{Gap}^{\text{F-U}}_{\text{Avg}}$ | $\text{SMIA}_{\text{AUC}}$ | $\text{Gap}^{\text{SMIA}}_{\text{Avg}}$ |
|---|---|---|---|---|---|---|---|---|
| **Forget 1%** | | | | | | | | |
| RETRAIN | $100.00_{\pm0.00}$ | $94.22_{\pm1.24}$ | $94.34_{\pm0.02}$ | $49.98_{\pm1.26}$ | $0.00_{\pm0.00}$ | $0.00_{\pm0.00}$ | $50.78_{\pm0.93}$ | $0.00_{\pm0.00}$ |
| BASE | $100.00_{\pm0.00}$ | $100.00_{\pm0.00}$ | $94.20_{\pm0.11}$ | $60.11_{\pm1.30}$ | $3.88_{\pm0.90}$ | $5.14_{\pm0.93}$ | $59.99_{\pm0.38}$ | $3.78_{\pm0.40}$ |
| NEGGRAD | $\mathbf{100.00}_{\pm0.00}$ | $100.00_{\pm0.00}$ | $\mathbf{94.17}_{\pm0.01}$ | $59.80_{\pm1.12}$ | $3.82_{\pm0.89}$ | $5.19_{\pm0.99}$ | $59.77_{\pm0.36}$ | $3.74_{\pm0.40}$ |
| NEGGRAD+ | $98.45_{\pm2.68}$ | $97.85_{\pm3.72}$ | $91.80_{\pm3.75}$ | $57.95_{\pm2.16}$ | $3.77_{\pm0.72}$ | $5.39_{\pm1.12}$ | $57.39_{\pm3.54}$ | $3.58_{\pm1.77}$ |
| FINETUNE | $97.44_{\pm0.49}$ | $\mathbf{94.67}_{\pm1.94}$ | $90.90_{\pm0.57}$ | $54.78_{\pm0.97}$ | $2.88_{\pm0.26}$ | $4.26_{\pm0.34}$ | $52.42_{\pm1.34}$ | $\underline{2.02}_{\pm0.73}$ |
| $\ell_1$-SPARSE | $97.15_{\pm0.77}$ | $\underline{94.89}_{\pm2.04}$ | $90.97_{\pm0.11}$ | $53.89_{\pm1.91}$ | $2.73_{\pm0.23}$ | $3.78_{\pm0.87}$ | $\underline{52.03}_{\pm0.33}$ | $2.04_{\pm0.67}$ |
| LUR | $74.07_{\pm4.72}$ | $73.63_{\pm5.13}$ | $72.21_{\pm4.39}$ | $\underline{50.51}_{\pm0.25}$ | $17.59_{\pm3.93}$ | $11.47_{\pm2.08}$ | $\mathbf{50.47}_{\pm0.41}$ | $17.24_{\pm2.10}$ |
| SSD | $\underline{99.95}_{\pm0.09}$ | $99.85_{\pm0.26}$ | $\underline{93.82}_{\pm0.68}$ | $59.69_{\pm1.14}$ | $3.84_{\pm0.88}$ | $5.28_{\pm1.04}$ | $59.69_{\pm0.75}$ | $3.78_{\pm0.47}$ |
| SALUN | $99.60_{\pm0.13}$ | $96.59_{\pm1.28}$ | $91.63_{\pm0.20}$ | $\mathbf{50.09}_{\pm3.34}$ | $\underline{1.64}_{\pm0.21}$ | $\underline{2.34}_{\pm0.16}$ | $48.40_{\pm3.01}$ | $\mathbf{1.97}_{\pm0.91}$ |
| AMUN | $99.28_{\pm0.16}$ | $87.85_{\pm1.89}$ | $91.84_{\pm0.34}$ | $44.17_{\pm1.49}$ | $3.94_{\pm1.56}$ | $3.90_{\pm1.33}$ | $41.65_{\pm1.95}$ | $4.68_{\pm0.79}$ |
| **ReGUn** | $99.37_{\pm0.50}$ | $95.33_{\pm1.94}$ | $90.93_{\pm1.14}$ | $48.90_{\pm0.51}$ | $\mathbf{1.21}_{\pm0.26}$ | $\mathbf{1.99}_{\pm0.44}$ | $47.06_{\pm0.38}$ | $2.22_{\pm0.70}$ |
| **Forget 10%** | | | | | | | | |
| RETRAIN | $100.00_{\pm0.00}$ | $94.28_{\pm0.38}$ | $93.81_{\pm0.19}$ | $50.19_{\pm0.92}$ | $0.00_{\pm0.00}$ | $0.00_{\pm0.00}$ | $49.86_{\pm0.29}$ | $0.00_{\pm0.00}$ |
| BASE | $100.00_{\pm0.00}$ | $100.00_{\pm0.00}$ | $94.29_{\pm0.13}$ | $59.50_{\pm0.89}$ | $3.88_{\pm0.29}$ | $4.90_{\pm0.43}$ | $59.51_{\pm0.44}$ | $3.96_{\pm0.17}$ |
| NEGGRAD | $\underline{99.90}_{\pm0.15}$ | $99.83_{\pm0.19}$ | $\mathbf{93.89}_{\pm0.39}$ | $59.47_{\pm0.78}$ | $3.82_{\pm0.29}$ | $4.81_{\pm0.44}$ | $58.91_{\pm0.14}$ | $3.70_{\pm0.18}$ |
| NEGGRAD+ | $99.85_{\pm0.13}$ | $99.26_{\pm0.53}$ | $93.02_{\pm0.65}$ | $59.10_{\pm0.74}$ | $3.71_{\pm0.33}$ | $4.85_{\pm0.37}$ | $57.69_{\pm0.75}$ | $3.44_{\pm0.31}$ |
| FINETUNE | $97.01_{\pm0.44}$ | $93.39_{\pm0.72}$ | $90.23_{\pm0.53}$ | $53.92_{\pm0.53}$ | $2.79_{\pm0.36}$ | $3.65_{\pm0.40}$ | $51.59_{\pm0.30}$ | $2.30_{\pm0.29}$ |
| $\ell_1$-SPARSE | $97.16_{\pm0.51}$ | $93.15_{\pm1.11}$ | $90.63_{\pm0.25}$ | $53.01_{\pm1.42}$ | $2.49_{\pm0.30}$ | $3.00_{\pm0.45}$ | $51.09_{\pm0.53}$ | $2.10_{\pm0.36}$ |
| LUR | $96.20_{\pm0.44}$ | $\underline{94.64}_{\pm0.30}$ | $90.56_{\pm0.40}$ | $54.27_{\pm0.36}$ | $2.90_{\pm0.26}$ | $3.66_{\pm0.62}$ | $52.37_{\pm0.26}$ | $2.48_{\pm0.22}$ |
| SSD | $\mathbf{100.00}_{\pm0.00}$ | $100.00_{\pm0.00}$ | $\underline{94.29}_{\pm0.13}$ | $59.50_{\pm0.89}$ | $3.88_{\pm0.30}$ | $4.90_{\pm0.44}$ | $59.51_{\pm0.44}$ | $3.96_{\pm0.17}$ |
| SALUN | $99.14_{\pm0.71}$ | $97.87_{\pm1.68}$ | $91.59_{\pm1.28}$ | $53.45_{\pm1.49}$ | $2.48_{\pm0.13}$ | $2.74_{\pm0.41}$ | $52.52_{\pm0.92}$ | $2.33_{\pm0.62}$ |
| AMUN | $99.29_{\pm0.20}$ | $\mathbf{94.63}_{\pm0.62}$ | $91.97_{\pm0.20}$ | $52.63_{\pm0.64}$ | $\mathbf{1.46}_{\pm0.15}$ | $\underline{2.14}_{\pm0.31}$ | $\mathbf{49.75}_{\pm0.48}$ | $\mathbf{0.75}_{\pm0.24}$ |
| **ReGUn** | $98.42_{\pm1.01}$ | $96.68_{\pm1.95}$ | $90.60_{\pm1.26}$ | $\mathbf{51.01}_{\pm0.64}$ | $\underline{2.00}_{\pm0.18}$ | $\mathbf{2.01}_{\pm0.67}$ | $\underline{50.55}_{\pm0.29}$ | $\underline{1.97}_{\pm0.65}$ |
| **Forget 50%** | | | | | | | | |
| RETRAIN | $100.00_{\pm0.00}$ | $90.89_{\pm0.52}$ | $90.31_{\pm0.41}$ | $50.24_{\pm0.36}$ | $0.00_{\pm0.00}$ | $0.00_{\pm0.00}$ | $50.25_{\pm0.29}$ | $0.00_{\pm0.00}$ |
| BASE | $100.00_{\pm0.00}$ | $100.00_{\pm0.00}$ | $94.17_{\pm0.02}$ | $56.42_{\pm0.32}$ | $4.79_{\pm0.30}$ | $5.02_{\pm0.35}$ | $59.39_{\pm0.47}$ | $5.53_{\pm0.22}$ |
| NEGGRAD | $\mathbf{100.00}_{\pm0.00}$ | $99.99_{\pm0.00}$ | $94.05_{\pm0.05}$ | $56.59_{\pm0.35}$ | $4.80_{\pm0.33}$ | $5.04_{\pm0.40}$ | $59.26_{\pm0.40}$ | $5.46_{\pm0.21}$ |
| NEGGRAD+ | $96.69_{\pm2.04}$ | $92.11_{\pm4.11}$ | $88.62_{\pm2.17}$ | $53.19_{\pm2.38}$ | $2.62_{\pm0.58}$ | $2.41_{\pm0.50}$ | $51.99_{\pm1.70}$ | $1.99_{\pm1.35}$ |
| FINETUNE | $95.86_{\pm1.04}$ | $90.60_{\pm0.98}$ | $88.10_{\pm0.46}$ | $52.40_{\pm0.62}$ | $2.39_{\pm0.47}$ | $2.19_{\pm0.05}$ | $51.27_{\pm0.06}$ | $1.92_{\pm0.42}$ |
| $\ell_1$-SPARSE | $96.86_{\pm0.86}$ | $91.87_{\pm1.32}$ | $88.82_{\pm0.75}$ | $52.81_{\pm0.69}$ | $2.09_{\pm0.10}$ | $2.03_{\pm0.26}$ | $51.38_{\pm0.47}$ | $1.69_{\pm0.49}$ |
| LUR | $96.76_{\pm1.30}$ | $\underline{91.15}_{\pm1.29}$ | $88.98_{\pm0.71}$ | $52.24_{\pm0.96}$ | $\underline{1.77}_{\pm0.11}$ | $1.66_{\pm0.18}$ | $\underline{51.13}_{\pm0.54}$ | $1.43_{\pm0.54}$ |
| SSD | $\underline{100.00}_{\pm0.00}$ | $100.00_{\pm0.00}$ | $94.18_{\pm0.01}$ | $56.42_{\pm0.32}$ | $4.79_{\pm0.30}$ | $5.02_{\pm0.34}$ | $59.39_{\pm0.47}$ | $5.53_{\pm0.22}$ |
| SALUN | $98.05_{\pm0.78}$ | $93.11_{\pm1.14}$ | $89.00_{\pm1.03}$ | $52.76_{\pm0.66}$ | $2.00_{\pm0.13}$ | $1.91_{\pm0.31}$ | $52.04_{\pm0.49}$ | $1.82_{\pm0.48}$ |
| AMUN | $97.60_{\pm1.64}$ | $\mathbf{90.97}_{\pm3.75}$ | $\underline{89.49}_{\pm1.96}$ | $\mathbf{51.02}_{\pm1.88}$ | $1.84_{\pm0.24}$ | $\underline{1.26}_{\pm0.52}$ | $\mathbf{50.06}_{\pm1.42}$ | $\mathbf{0.87}_{\pm1.20}$ |
| **ReGUn** | $98.77_{\pm0.20}$ | $93.44_{\pm0.36}$ | $\mathbf{90.11}_{\pm0.12}$ | $\underline{52.10}_{\pm0.11}$ | $\mathbf{1.48}_{\pm0.08}$ | $\mathbf{1.07}_{\pm0.17}$ | $51.50_{\pm0.41}$ | $\underline{1.31}_{\pm0.23}$ |

Table 1: **Forgetting and utility results for ResNet-18 on CIFAR-10.** Studied under 1%, 10%, and 50% random forgetting. RA/FA/TA denote retain, forget, and test accuracy; $\text{RMIA}_{\text{AUC}}$ and $\text{SMIA}_{\text{AUC}}$ report membership inference performance, with 50% corresponding to chance-level discrimination. Gap metrics measure deviation from RETRAIN, so lower is better. Results are mean $\pm$ std over three seeds, with all values in percent. **Bold** and underlined denote best and second best, where "best" denotes smallest gap to RETRAIN.

| | RA | FA | TA | $\text{RMIA}_{\text{AUC}}$ | $\text{Gap}_{\text{Avg}}$ | $\text{Gap}_{\text{Avg}}^{\text{F-U}}$ | $\text{SMIA}_{\text{AUC}}$ | $\text{Gap}_{\text{Avg}}^{\text{SMIA}}$ |
|---|---|---|---|---|---|---|---|---|
| **Forget 1%** | | | | | | | | |
| Retrain | 99.99 ±0.00 | 61.48 ±1.23 | 60.89 ±0.15 | 49.79 ±1.38 | 0.00 ±0.00 | 0.00 ±0.00 | 50.50 ±0.69 | 0.00 ±0.00 |
| Base | 99.99 ±0.00 | 99.96 ±0.06 | 61.24 ±0.07 | 87.77 ±0.18 | 19.21 ±0.41 | 19.15 ±0.69 | 94.68 ±0.10 | 20.75 ±0.36 |
| NegGrad | **99.99** ±0.00 | 99.96 ±0.06 | **61.22** ±0.06 | 87.78 ±0.18 | 19.20 ±0.42 | 19.15 ±0.72 | 94.68 ±0.09 | 20.75 ±0.36 |
| NegGrad+ | 94.31 ±0.13 | 68.78 ±1.47 | 48.99 ±0.31 | 66.99 ±1.96 | 10.52 ±0.59 | 14.54 ±0.98 | 62.44 ±1.16 | 9.21 ±0.59 |
| Finetune | 97.11 ±0.19 | 63.70 ±1.74 | 52.31 ±0.36 | 62.48 ±0.22 | 6.60 ±0.59 | 10.63 ±0.92 | 57.26 ±0.66 | 5.11 ±0.59 |
| $\ell_1$-sparse | 96.49 ±0.21 | **61.26** ±1.41 | 51.94 ±0.03 | 62.37 ±0.96 | 6.54 ±0.33 | 10.76 ±1.02 | 56.86 ±1.06 | 4.76 ±0.57 |
| LUR | 43.35 ±0.21 | 40.30 ±1.45 | 38.24 ±0.21 | 51.98 ±0.46 | 25.65 ±0.32 | 12.41 ±0.52 | **51.66** ±0.90 | 25.41 ±0.56 |
| SSD | 82.26 ±16.32 | 67.74 ±17.77 | 41.99 ±7.57 | 68.08 ±4.21 | 17.63 ±4.28 | 18.59 ±1.07 | 67.13 ±4.30 | 14.88 ±6.42 |
| SalUn | 99.30 ±0.06 | 63.00 ±2.12 | 53.16 ±0.69 | 46.36 ±1.39 | **3.73** ±0.20 | 5.59 ±0.80 | 42.57 ±0.38 | **4.47** ±0.67 |
| Amun | 98.60 ±0.15 | 46.44 ±0.58 | 52.59 ±0.46 | **51.03** ±0.57 | 6.55 ±0.62 | **4.91** ±0.48 | 46.31 ±0.28 | 7.23 ±0.41 |
| **ReGUn** | 98.64 ±0.13 | 52.89 ±0.89 | 52.26 ±0.40 | 45.07 ±0.83 | 5.82 ±0.16 | 6.68 ±0.42 | 40.50 ±0.13 | 7.14 ±0.43 |
| **Forget 10%** | | | | | | | | |
| Retrain | 99.99 ±0.00 | 59.54 ±0.54 | 59.27 ±0.30 | 50.30 ±0.66 | 0.00 ±0.00 | 0.00 ±0.00 | 50.35 ±0.00 | 0.00 ±0.00 |
| Base | 99.98 ±0.00 | 99.98 ±0.01 | 61.03 ±0.23 | 86.40 ±0.26 | 19.58 ±0.32 | 18.93 ±0.61 | 94.70 ±0.00 | 21.64 ±0.16 |
| NegGrad | **99.98** ±0.00 | 99.98 ±0.01 | **61.02** ±0.22 | 86.43 ±0.26 | 19.58 ±0.32 | 18.94 ±0.60 | 94.72 ±0.03 | 21.64 ±0.16 |
| NegGrad+ | 89.85 ±0.55 | 53.03 ±0.69 | 46.49 ±0.31 | 57.18 ±0.11 | 9.07 ±0.48 | 9.83 ±0.48 | 54.47 ±0.41 | 8.39 ±0.30 |
| Finetune | 97.07 ±0.45 | 61.54 ±0.53 | 51.00 ±0.75 | 62.40 ±0.34 | 6.32 ±0.18 | 10.18 ±0.11 | 56.86 ±0.17 | 4.93 ±0.30 |
| $\ell_1$-sparse | 95.61 ±1.09 | **58.92** ±2.72 | 50.26 ±0.33 | 61.00 ±2.48 | 6.39 ±0.24 | 9.85 ±1.40 | 56.12 ±1.84 | 4.95 ±0.88 |
| LUR | 71.03 ±0.51 | 58.42 ±1.01 | 52.84 ±0.11 | 56.06 ±0.03 | 10.56 ±0.25 | 6.09 ±0.28 | 53.82 ±0.59 | 10.00 ±0.36 |
| SSD | 99.16 ±1.05 | 98.89 ±1.31 | 55.87 ±3.84 | 84.36 ±0.15 | 19.41 ±0.85 | 18.73 ±1.95 | 86.89 ±2.45 | 20.03 ±1.22 |
| SalUn | 91.53 ±0.67 | 64.36 ±0.80 | 49.88 ±0.24 | 55.77 ±0.80 | 7.03 ±0.24 | 7.43 ±0.31 | 52.21 ±0.50 | 6.13 ±0.33 |
| Amun | 96.04 ±0.51 | 53.01 ±0.28 | 51.06 ±0.39 | 55.35 ±0.30 | 5.93 ±0.15 | 6.63 ±0.36 | **51.92** ±0.26 | 5.07 ±0.24 |
| **ReGUn** | 98.98 ±0.15 | 63.30 ±1.38 | 52.72 ±0.25 | **49.86** ±0.83 | **3.05** ±0.21 | **3.70** ±0.54 | 45.54 ±0.09 | **4.03** ±0.39 |
| **Forget 50%** | | | | | | | | |
| Retrain | 99.99 ±0.00 | 48.34 ±0.20 | 47.95 ±0.12 | 50.30 ±0.19 | 0.00 ±0.00 | 0.00 ±0.00 | 50.24 ±0.00 | 0.00 ±0.00 |
| Base | 99.99 ±0.00 | 99.99 ±0.00 | 61.20 ±0.20 | 79.74 ±0.05 | 23.58 ±0.04 | 21.34 ±0.03 | 94.70 ±0.00 | 27.34 ±0.08 |
| NegGrad | **99.99** ±0.00 | 99.99 ±0.00 | 61.19 ±0.16 | 79.84 ±0.05 | 23.61 ±0.05 | 21.39 ±0.05 | 94.61 ±0.13 | 27.32 ±0.08 |
| NegGrad+ | 96.71 ±0.58 | 48.92 ±2.99 | 43.63 ±0.98 | 56.66 ±2.25 | 4.12 ±0.18 | 5.34 ±0.70 | 54.14 ±1.32 | 3.02 ±0.87 |
| Finetune | 97.35 ±0.43 | 55.11 ±1.17 | 45.74 ±0.82 | 61.46 ±0.42 | 5.70 ±0.10 | 6.69 ±0.27 | 56.78 ±0.21 | 4.54 ±0.38 |
| $\ell_1$-sparse | 94.25 ±0.27 | 47.29 ±0.32 | 43.48 ±0.53 | 55.23 ±0.35 | 4.05 ±0.30 | 4.70 ±0.27 | 53.12 ±0.04 | 3.54 ±0.18 |
| LUR | 98.47 ±0.15 | 43.60 ±0.24 | 46.33 ±0.39 | 44.91 ±0.42 | 3.32 ±0.06 | 3.50 ±0.21 | 47.08 ±0.22 | 2.76 ±0.14 |
| SSD | 77.45 ±7.00 | 75.75 ±7.79 | 38.10 ±2.80 | 74.84 ±3.49 | 21.09 ±0.59 | 17.19 ±0.91 | 74.77 ±2.40 | 21.08 ±2.78 |
| SalUn | 97.79 ±1.69 | 59.57 ±0.51 | **47.77** ±0.30 | 58.04 ±0.95 | 5.36 ±0.05 | 4.01 ±0.43 | 55.38 ±0.13 | 4.69 ±0.45 |
| Amun | 95.47 ±0.57 | 55.48 ±0.83 | 48.21 ±0.75 | 59.37 ±0.34 | 5.34 ±0.15 | 4.86 ±0.13 | 55.33 ±0.10 | 4.25 ±0.32 |
| **ReGUn** | 99.97 ±0.01 | **48.21** ±0.64 | 45.57 ±0.44 | **47.88** ±0.19 | **1.37** ±0.09 | 2.40 ±0.19 | **48.16** ±0.42 | **1.15** ±0.23 |

Table 2: **Forgetting and utility results for Swin-T on Tiny-ImageNet.** Studied under 1%, 10%, and 50% random forgetting. RA/FA/TA denote retain, forget, and test accuracy; $\text{RMIA}_{\text{AUC}}$ and $\text{SMIA}_{\text{AUC}}$ report membership inference performance, with 50% corresponding to chance-level discrimination. Gap metrics measure deviation from Retrain, so lower is better. Results are mean ± std over three seeds, with all values in percent. **Bold** and underlined denote best and second best, where "best" denotes smallest gap to Retrain.

## 5 Results

To evaluate whether indistinguishability-based supervision in REGUN provides an effective signal for approximate machine unlearning, we address five questions: *(i) Unseen behavior as supervision:* Does a held-out reference distribution improve unlearning over degradation-based objectives? *(ii) Forgetting–utility trade-off:* Does REGUN better balance forgetting efficacy and retained utility? *(iii) Computational cost:* What overhead does reference-guided unlearning introduce? *(iv) Reference construction:* How do the teacher model and held-out selector affect performance? *(v) Held-out data dependence:* How sensitive is REGUN to the size of the held-out supervision set?

**Unseen Behavior is a Strong Unlearning Signal.** Tables 1 and 2 present our main findings across the two main-text settings and three forget fractions. Overall, REGUN achieves the strongest average agreement with the retrain-from-scratch baseline in most settings, indicating a favorable forgetting–utility trade-off. In the ResNet-18/CIFAR-10 setting, REGUN obtains the best $\text{Gap}_{\text{Avg}}$ in the 1% and 50% forgetting regimes and remains competitive at 10%, where it is the runner-up. This improvement is mainly driven by reduced membership inference risk relative to the BASE model, while maintaining competitive test and forget accuracy. Although REGUN is not always the best on each individual submetric, this is expected in a trade-off setting: different baselines favor different aspects of retrain matching, whereas REGUN is consistently among the closest methods in aggregate. The advantage becomes clearer in the higher-resolution Transformer setting with Swin-T on Tiny-ImageNet. Here, REGUN achieves the best $\text{Gap}_{\text{Avg}}$ in the 10% and 50% forgetting regimes and remains the runner-up at 1%, where the forget signal is weaker and several methods perform similarly. More broadly, all approximate methods exhibit larger gaps to RETRAIN in the Swin-T setting than in the ResNet-18 setting, suggesting that Transformer-based unlearning remains a more challenging and comparatively underexplored regime. Among the baselines, performance varies substantially: specialized methods such as AMUN and SALUN are often competitive, while simpler methods such as FINETUNE and NEGGRAD+ can remain strong in individual metrics in some Transformer settings. Results on the two additional settings in Appendix B.2 follow the same general trend: REGUN tends to perform best in aggregate, while the strongest method on individual metrics varies across datasets and forget fractions. Taken together, these results suggest that reference-guided distillation provides a strong and stable unlearning signal without relying on aggressive loss-ascent updates to approach RETRAIN behavior.

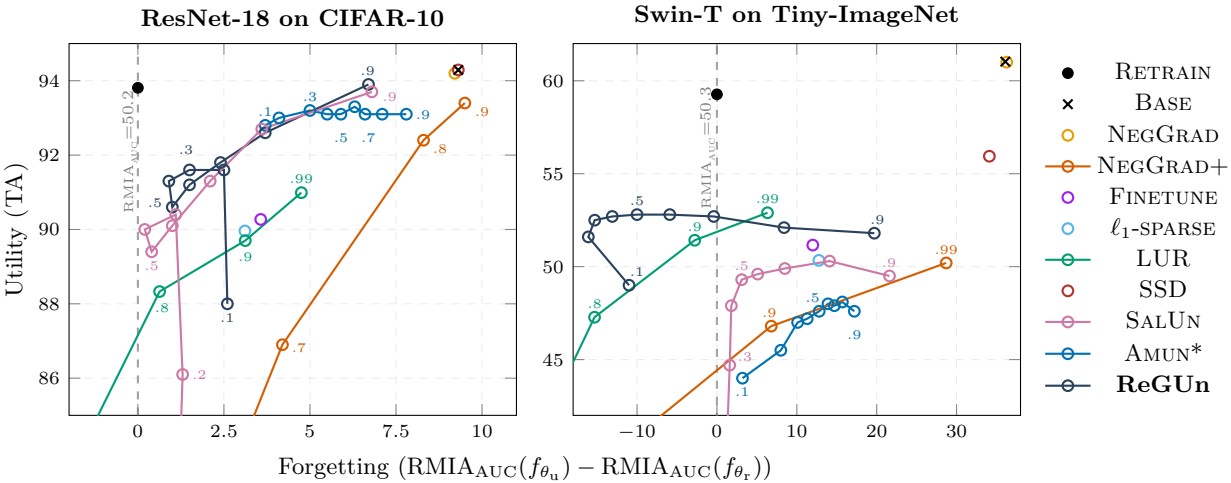

Figure 2: **Forgetting–utility trade-offs.** Shown for ResNet-18 on CIFAR-10 and Swin-T on Tiny-ImageNet under random 10% forgetting. The $y$-axis reports test accuracy and the $x$-axis the $\text{RMIA}_{\text{AUC}}$ gap to the retrain-from-scratch baseline; higher TA and proximity to the dashed vertical line is better. Where supported, we sweep $w \in [0.1, 0.9]$ in the retain–forget objective, with an additional $w = 0.99$ for NEGGRAD+ and LUR. Points are labeled by $w$; * marks variants reparameterized from the original loss to sweep $w$.

**Understanding the Forgetting–Utility Trade-off.** Figure 2 visualizes the trade-off between retained utility and forgetting behavior. The $y$-axis reports test accuracy, while the $x$-axis reports the gap between each method's $\text{RMIA}_{\text{AUC}}$ and the retrain-from-scratch baseline; therefore, points closer to the dashed vertical line and higher on the plot are preferable. For methods that admit an explicit retain–forget trade-off, we reparameterize the objective as $(1 - w) \cdot \mathcal{L}_{\text{forget}} + w \cdot \mathcal{L}_{\text{retain}}$ and sweep $w \in [0.1, 0.9]$ to examine how each method behaves as the retain–forget balance is varied. In the ResNet-18/CIFAR-10 setting, several methods occupy a similar region of the plot, consistent with the main-table observation that this benchmark is close to saturated. The additional ResNet-18 results in Figure 5 show a similar pattern on CIFAR-100, where several baselines reach comparable regions of the trade-off space, while the ResNet-18/Tiny-ImageNet setting shows a somewhat clearer advantage for REGUN. More prominently, the Swin-T/Tiny-ImageNet setting reveals the clearest separation: many baselines improve forgetting only at the cost of lower test accuracy, whereas REGUN maintains relatively stable utility while remaining close to the retrain $\text{RMIA}_{\text{AUC}}$ level. This suggests that reference-guided distillation provides a less destructive forgetting signal, improving the forgetting–utility trade-off without relying on aggressive loss-ascent behavior.

**Computational Cost of ReGUn.** We briefly analyze REGUN's runtime in terms of leading computational operations. A full method-wise comparison with empirical runtime results is given in Appendix A.2. Let $C_G(b)$ denote the cost of one gradient update on a batch of size $b$, and let $C_F(b)$ denote the cost of a no-gradient forward pass. Each REGUN step performs one standard first-order update on the retain and forget minibatches of sizes $b_{\text{r}}$ and $b_{\text{f}}$, and additionally constructs the reference target from a held-out minibatch of size $b_{\text{h}}$. Its cost over $T$ unlearning steps is $\mathcal{T}_{\text{REGUN}} = T [C_G(b_{\text{f}} + b_{\text{r}}) + C_F(b_{\text{h}}) + \mathcal{O}(b_{\text{h}}K)]$, with $b_{\text{h}} = b_{\text{f}}$ by default and $K$ denoting the number of classes. The first term is the standard gradient update through the current model, while the second term is the no-gradient reference-model forward pass used to construct the held-out target. The final term accounts for averaging the held-out probability vectors into the batch-level reference distribution. Thus, relative to a standard retain–forget first-order update such as NEGGRAD+, the only additional leading operation is one no-gradient forward pass through the frozen reference model. If the reference predictions $p_\phi(\cdot \mid x)$ for $x \in \mathcal{D}_{\text{h}}$ are precomputed, this overhead reduces to sampling and averaging cached $K$-dimensional probability vectors, with one-time preprocessing cost $C_F(n_{\text{h}})$ and storage $\mathcal{O}(n_{\text{h}}K)$. Overall, REGUN remains close to standard first-order unlearning methods in leading-order complexity: its main extra cost is reference construction, rather than importance-estimation passes, saliency-mask construction, adversarial-example generation, or inner-loop computation.

| | Forget 1% | | | Forget 10% | | | Forget 50% | | |
|---|---|---|---|---|---|---|---|---|---|
| | TA | $\text{RMIA}_{\text{AUC}}$ | $\text{Gap}_{\text{Avg}}$ | TA | $\text{RMIA}_{\text{AUC}}$ | $\text{Gap}_{\text{Avg}}$ | TA | $\text{RMIA}_{\text{AUC}}$ | $\text{Gap}_{\text{Avg}}$ |
| **ResNet-18 on CIFAR-10** | | | | | | | | | |
| Oracle Retrained | 91.95 ±0.03 | 51.08 ±1.20 | 1.70 ±0.65 | 90.45 ±0.63 | **50.45** ±0.91 | 2.02 ±0.15 | 89.45 ±0.61 | 52.19 ±0.12 | **1.54** ±0.26 |
| Online Student | 91.83 ±0.15 | **50.53** ±1.15 | **1.47** ±0.67 | 89.83 ±1.72 | 50.95 ±0.73 | 2.28 ±0.28 | 88.62 ±0.60 | 52.13 ±0.33 | 1.74 ±0.19 |
| EMA 0.99 | 91.98 ±0.09 | 50.78 ±1.07 | 1.56 ±0.62 | 89.43 ±1.62 | 50.70 ±0.85 | 2.41 ±0.41 | 89.04 ±0.77 | 52.18 ±0.05 | 1.62 ±0.07 |
| EMA 0.999 | 91.99 ±0.07 | 50.77 ±1.08 | 1.58 ±0.64 | 88.77 ±2.32 | 50.78 ±0.85 | 2.85 ±1.06 | 88.39 ±0.65 | 52.17 ±0.55 | 1.72 ±0.10 |
| Frozen Base | **92.15** ±0.17 | 50.71 ±0.95 | 1.49 ±0.81 | 90.63 ±0.96 | **50.46** ±1.05 | **1.99** ±0.26 | 88.81 ±0.82 | **52.07** ±0.25 | 1.64 ±0.15 |
| **Swin-T on Tiny-ImageNet** | | | | | | | | | |
| Oracle Retrained | 51.24 ±1.03 | 57.24 ±1.67 | 5.76 ±0.42 | 52.06 ±0.89 | **50.04** ±0.77 | **3.00** ±0.22 | 45.82 ±0.33 | 47.59 ±0.45 | 1.36 ±0.09 |
| Online Student | 51.23 ±0.88 | 56.15 ±0.28 | 4.75 ±0.33 | 52.43 ±0.24 | 51.01 ±1.10 | 3.23 ±0.61 | 44.96 ±0.59 | 47.49 ±0.77 | 1.79 ±0.08 |
| EMA 0.99 | 51.61 ±0.39 | 56.89 ±0.40 | 5.06 ±0.04 | 52.41 ±0.33 | 50.82 ±1.02 | 3.08 ±0.58 | 45.29 ±0.51 | 47.77 ±0.33 | 1.54 ±0.23 |
| EMA 0.999 | **51.71** ±0.34 | 56.78 ±0.46 | 4.90 ±0.07 | 52.45 ±0.23 | 50.79 ±1.09 | 3.10 ±0.48 | 45.22 ±0.48 | **48.03** ±0.32 | 1.53 ±0.12 |
| Frozen Base | 51.23 ±0.23 | **56.07** ±0.84 | **4.68** ±0.47 | **52.66** ±0.20 | 49.88 ±0.81 | **3.00** ±0.21 | 45.64 ±0.40 | 47.86 ±0.18 | **1.34** ±0.13 |

Table 3: **Teacher ablation results. Bold** and underlined denote best and second best, where "best" denotes smallest gap to RETRAIN.

**The Effect of the Teacher: A Frozen Base Model is a Strong Reference Teacher.** REGUN uses the original trained model $f_{\theta_0}$ as a frozen reference teacher to construct the held-out target distribution. A natural question is whether this simple choice is sufficient, or whether the reference should instead be produced by a teacher that evolves during unlearning or more closely approximates the retrained oracle. We therefore compare the frozen base teacher with four alternatives: the online student $f_\theta$, whose reference

distribution changes throughout unlearning; two EMA teachers, $f_{\text{EMA}}$ with decay factors 0.99 and 0.999; and an oracle retrained teacher $f_{\theta_{\text{r}}}$ trained only on the retain set. The oracle represents the ideal post-unlearning model, but is unavailable in realistic approximate unlearning settings. Table 3 shows that the frozen base model is already a strong reference teacher. Across both main settings, it is consistently competitive with online, EMA, and oracle retrained teachers, and even achieves the best or tied-best $\text{Gap}_{\text{Avg}}$ in several regimes. Notably, the oracle retrained teacher does not consistently outperform the frozen base model, despite being trained on the ideal retain-only data. The full metrics in Tables 5 and 6 support the same conclusion. The frozen base teacher remains best or near-best across utility, forgetting, and membership-inference metrics. Together, these results indicate that REGUN does not require an adaptive or oracle-like teacher. What matters most is the reference construction; the teacher is queried only on disjoint held-out non-member examples, and its predictions are aggregated into a class-conditioned target.

| | Forget 1% | | | Forget 10% | | | Forget 50% | | |
|---|---|---|---|---|---|---|---|---|---|
| | TA | $\text{RMIA}_{\text{AUC}}$ | $\text{Gap}_{\text{Avg}}$ | TA | $\text{RMIA}_{\text{AUC}}$ | $\text{Gap}_{\text{Avg}}$ | TA | $\text{RMIA}_{\text{AUC}}$ | $\text{Gap}_{\text{Avg}}$ |
| **ResNet-18 on CIFAR-10** | | | | | | | | | |
| Uniform | 91.97 ±0.06 | 50.74 ±0.93 | 1.52 ±0.31 | 88.93 ±3.30 | **50.44** ±0.69 | 2.90 ±1.57 | 88.62 ±1.01 | **51.91** ±0.38 | 1.64 ±0.22 |
| Feature-1NN | 91.95 ±0.15 | 50.80 ±0.92 | 1.55 ±0.50 | 89.99 ±1.47 | **50.44** ±0.68 | 2.13 ±0.27 | 88.63 ±0.14 | 51.92 ±0.16 | **1.61** ±0.12 |
| Class-Conditioned | **92.15** ±0.17 | 50.71 ±0.95 | 1.49 ±0.81 | 90.63 ±0.96 | 50.46 ±1.05 | **1.99** ±0.26 | **88.81** ±0.82 | 52.07 ±0.25 | 1.64 ±0.15 |
| **Swin-T on Tiny-ImageNet** | | | | | | | | | |
| Uniform | 50.83 ±0.48 | **55.77** ±0.61 | 4.88 ±0.15 | 52.36 ±0.54 | 50.50 ±0.73 | 3.06 ±0.35 | 44.88 ±0.77 | 47.42 ±0.18 | 1.82 ±0.41 |
| Feature-1NN | 51.07 ±0.81 | 56.56 ±2.06 | 5.35 ±0.29 | **52.72** ±0.83 | 50.95 ±0.83 | 3.33 ±0.35 | **45.72** ±0.50 | **48.27** ±0.36 | 1.36 ±0.18 |
| Class-Conditioned | **51.23** ±0.23 | 56.07 ±0.84 | **4.68** ±0.47 | 52.66 ±0.20 | 49.88 ±0.81 | **3.00** ±0.21 | 45.64 ±0.40 | 47.86 ±0.18 | **1.34** ±0.13 |

Table 4: **Selector ablation results. Bold** and underlined denote best and second best, where "best" denotes smallest gap to RETRAIN.

**Class-Conditioned References Improve Unlearning.** REGUN constructs its reference distribution by selecting held-out samples from $\mathcal{D}_{\text{h}}$. We compare the default class-conditioned selector, which matches the class histogram of the forget batch, against uniform sampling (Algorithm 3) and feature-nearest-neighbor retrieval (Algorithm 4). As shown in Table 4, selector choice has a relatively small effect at lower forget fractions, where all strategies yield similar performance. However, class-conditioned sampling is consistently competitive and becomes more beneficial as the forget fraction increases. On ResNet-18/CIFAR-10, it achieves the best $\text{Gap}_{\text{Avg}}$ at 1% and 10% forgetting and remains comparable at 50%. On Swin-T/Tiny-ImageNet, it obtains the best $\text{Gap}_{\text{Avg}}$ at 1% and 50% forgetting, and matches the best result at 10%. Overall, these results support the use of class-conditioned references as the default selector. Matching the held-out batch to the forget-batch class histogram preserves label-prior structure in the target distribution, yielding a more stable non-member reference than class-agnostic sampling, especially when stronger forgetting is required.

**ReGUn is Stable Across Held-Out Set Sizes.** We next analyze the sensitivity of REGUN to the amount of held-out supervision $\mathcal{D}_{\text{h}}$ used to construct the reference distribution. Figure 3 reports $\text{Gap}_{\text{Avg}}$ when progressively subsampling $\mathcal{D}_{\text{h}}$ on varying forgetting regimes. On ResNet-18/CIFAR-10, performance is highly stable across held-out fractions; the average gap varies only mildly, even when the held-out split is reduced by several orders of magnitude. This suggests that, in simpler low-class-count settings, a very small number of held-out non-member examples can already provide an effective reference signal. In contrast, the Swin-T/Tiny-ImageNet setting shows a clearer dependence on held-out diversity, especially at the smallest subsample fraction. However, even these highly reduced held-out subsets still yield meaningful unlearning behavior. Performance improves once the held-out fraction reaches $10^{-2}$ and then remains relatively stable as more held-out data is added, suggesting that only a modest amount of class coverage is needed to recover most of the benefit of the full held-out split. This behavior is consistent with the known data dependence of Transformer-based architectures, which typically require stronger supervision signals to maintain stable representations. Overall, this ablation shows that REGUN does not rely on a large auxiliary dataset. Held-out data is used primarily to estimate class-conditioned non-member behavior, not to retrain the model. Small held-out subsets are already sufficient in simpler settings, while more diverse held-out coverage provides additional stability as the dataset, number of classes, and backbone complexity increase.

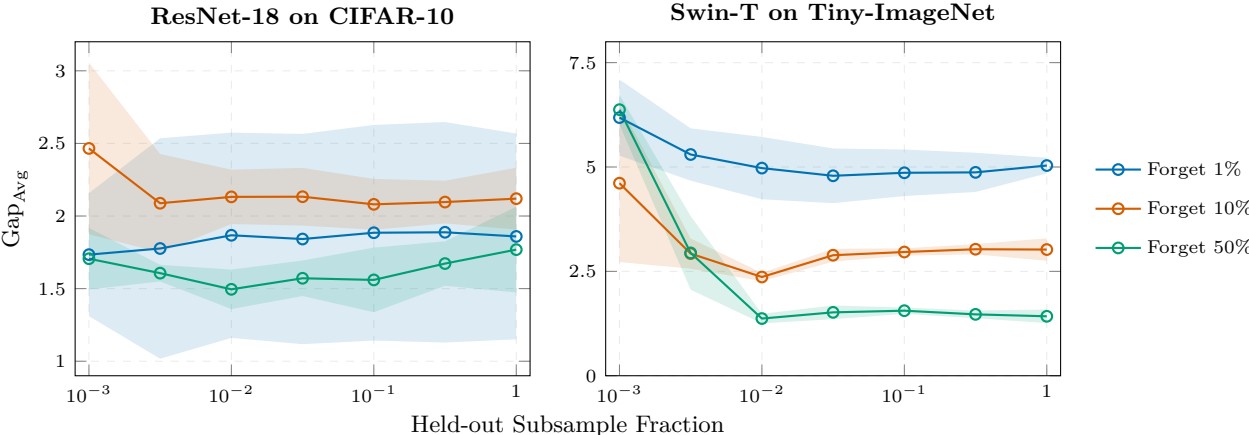

Figure 3: **Held-out subsample ablation.** $\text{Gap}_{\text{Avg}}$ is reported as a function of the fraction of the held-out split used to construct REGUN reference distributions. A fraction of 1 uses the full held-out split, corresponding to $\approx 9\%$ of the original training set $\mathcal{D}_{\text{orig}}$, or about 4,500 images for CIFAR-10 and 9,000 for Tiny-ImageNet. Lines show the mean across seeds; shaded regions indicate $\pm 1$ sample standard deviation.

## 6 Conclusion

We introduced REGUN, a machine unlearning framework for vision that reframes approximate unlearning from performance degradation to distributional matching. Instead of pushing the model to be wrong on forget examples via often poorly conditioned objectives, REGUN aligns the model's behavior on forget samples with its predictive distribution on disjoint held-out, non-member data. This design is motivated by the principle of indistinguishability: a truly unlearned model should treat the forget set as if it were a future unseen test set, precisely the property that many membership inference attacks aim to detect. Across diverse architectures, model scales, datasets, and forgetting regimes, our results show that held-out reference supervision is a simple yet effective unlearning signal. In particular, it achieves a favorable forgetting–utility trade-off, performs strongly under membership inference evaluation, and remains computationally on par with standard fine-tuning-based unlearning. Taken together, these findings suggest that approximate unlearning can benefit substantially from aligning the optimization objective with the evaluation criterion: making forgotten data behave like non-member data. We show that simple procedures can outperform substantially more complex methods when optimized toward the right objective.

**Limitations and Future Work.** A practical consideration for REGUN is that it relies on a labeled held-out set, which may not always be available in real-world deployments. While our results suggest that REGUN can remain effective with relatively small held-out subsets, it still assumes access to labeled held-out data from a distribution sufficiently aligned with the training data. Relaxing this requirement is an interesting direction for future work, for example, by exploring references from external datasets, out-of-distribution data, or synthetic samples. Beyond this, the flexibility of this framework opens several avenues for research. While we focused primarily on class-conditioned references, future variants of REGUN could explore richer semantic conditioning strategies, such as adaptive instance-conditioned references, or learned reference selection mechanisms. Beyond discriminative tasks, extending these principles to generative modeling remains a critical frontier, particularly for understanding how distributional matching objectives scale to high-dimensional foundation models. REGUN is not intended to provide a formal unlearning guarantee. Rather, it is an approximate unlearning procedure designed to align forget-set behavior with held-out, non-member behavior. Its effectiveness is therefore assessed empirically, serving as a practical proxy for indistinguishability rather than formal certificates. Overall, our results show that even simple unlearning objectives can be highly effective when aligned with the right principle: making forgotten data behave like unseen data. We hope these findings encourage future research to prioritize indistinguishability as a core objective, even when developing more complex unlearning solutions.

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

# A  Experimental Protocol and Computational Cost

## A.1  Implementation and Hyperparameter Details for ReGUn and Baselines

We summarize implementation details and hyperparameter search spaces and choices below. For additional implementation details, please refer to the published code (Footnote 1).

**Base & Retrain.**  For base training (from scratch) and retraining, we follow standard recipes. For ResNet-18, we train for 100 epochs using SGD with momentum 0.9, learning rate 0.1, batch size 128, and apply standard data augmentation consisting of random crop with padding (32×32, pad=4), random horizontal flipping (p=0.5), and mild color jitter (brightness/contrast/saturation=0.1, hue=0.02). For Swin-T, we train for 200 epochs using AdamW with learning rate $3 \cdot 10^{-4}$ and a cosine schedule, batch size 128, and apply the standard data augmentation consisting of random horizontal flipping (p=0.5), and mild color jitter (brightness/contrast/saturation=0.1, hue=0.02) plus additional stronger augmentation in the form of RandAugment (N=2, M=9) and Random Erasing (p=0.25, area 2–20%). For ResNet-18, we use the native data resolutions of 32×32 on CIFAR, and 64×64 on Tiny-ImageNet. For Swin-T on Tiny-ImageNet, we use 224×224 resolution to align with the architecture's default patch-based configuration.

For unlearning (except NEGGRAD), we use a fixed budget of 10 epochs for ResNet-18 and 20 epochs for Swin-T. All other settings are kept the same as base training except that we omit the strong data augmentation in the Swin-T setup. The detailed configurations per method are the following:

**NegGrad.**  We run gradient ascent for 2 epochs and tune the learning rate. ResNet-18: lr ∈ {5e−2, 1e−2, 5e−3, 1e−3}. Swin-T: lr ∈ {1e−5, 5e−6, 1e−6, 5e−7, 1e−7, 5e−8, 1e−8}.

**NegGrad+.**  We combine a negative-gradient forget objective with a retain objective weighted by $w$. We tune the learning rate and the retain weight $w$. ResNet-18: lr ∈ {1e−1, 5e−2, 1e−2, 5e−3}, $w \in [0.8, 0.99]$. Swin-T: lr ∈ {1e−3, 5e−4, 1e−4}, $w \in [0.8, 0.99]$.

**Finetune.**  We fine-tune on the full retain set $\mathcal{D}_r$ and tune the learning rate. ResNet-18: lr ∈ {1e−1, 5e−2, 1e−2, 5e−3}. Swin-T: lr ∈ {1e−3, 5e−4, 1e−4}.

**$\ell_1$-sparse.**  We implement sparsity-aware unlearning as fine-tuning on $\mathcal{D}_r$ with an $\ell_1$ penalty weight $\gamma$. We tune the learning rate and $\gamma$. ResNet-18: lr ∈ {1e−1, 5e−2, 1e−2, 5e−3}, $\gamma \in [5e−6, 5e−3]$. Swin-T: lr ∈ {1e−3, 5e−4, 1e−4}, $\gamma \in [5e−7, 5e−5]$.

**LUR.**  We follow the original LUR setup and tune the learning rate and retain weight $w$. We use the LUR pruning/reinitialization step before unlearning, with batch-normalization parameters excluded from pruning, and fix the inner-step size to $\alpha = 10^{-2}$. ResNet-18: lr ∈ {1e−1, 5e−2, 1e−2, 5e−3}, $w \in [0.1, 0.99]$. Swin-T: lr ∈ {1e−3, 5e−4, 1e−4}, $w \in [0.1, 0.99]$. Since full-model LUR is not computationally practical for larger models such as Swin-T at $224 \times 224$ resolution in our setup, for Swin-T experiments, we restrict unlearning and pruning to the last stage, final normalization layer, and classification head.

**SSD.**  For ResNet-18, we use the recommended settings from the paper ($\alpha = 10.0$, $\lambda = 1.0$). For Swin-T, we tune $\alpha$ and $\lambda$. Swin-T: $\alpha \in [1, 10]$, $\lambda \in [0.7, 1.0]$. Note that SSD is not designed for large random-forgetting regimes, which should be considered when interpreting the results.

**SalUn.**  We fix the sparsity threshold to 50% and tune the learning rate and a retain weight $w$ that balances the retain objective with the forgetting loss. ResNet-18: lr ∈ {1e−1, 5e−2, 1e−2, 5e−3}, $w \in [0.1, 0.9]$. Swin-T: lr ∈ {1e−3, 5e−4, 1e−4}, $w \in [0.1, 0.9]$.

**Amun.**  Let $\mathcal{D}_A$ denote the adversarial examples generated from the forget data. We evaluate all four configurations described in the paper, corresponding to fine-tuning on $\mathcal{D}_A \cup \mathcal{D}_f \cup \mathcal{D}_r$, $\mathcal{D}_A \cup \mathcal{D}_f$, $\mathcal{D}_A \cup \mathcal{D}_r$, and $\mathcal{D}_A$ and tune the learning rate. ResNet-18: lr ∈ {1e−1, 5e−2, 1e−2, 5e−3}. Swin-T: lr ∈ {1e−3, 5e−4, 1e−4}.

**ReGUn.** We tune the learning rate and the retain/forget trade-off weight $w$ (corresponding to $\lambda_\mathrm{r} = w$ and $\lambda_\mathrm{f} = 1 - w$ in the main objective). ResNet-18: lr $\in \{1e{-}1, 5e{-}2, 1e{-}2, 5e{-}3\}$, $w \in [0.1, 0.9]$. Swin-T: lr $\in \{1e{-}3, 5e{-}4, 1e{-}4\}$, $w \in [0.1, 0.9]$.

To produce the results for Figure 2, for each method, we fixed the best hyperparameter setting (excluding $w$) and then swept $w$. NEGGRAD+, SALUN, LUR, and REGUN inherently support this sweep. For AMUN, we rewrote the objective as $(1 - w)\mathcal{L}_\mathrm{forget} + w\mathcal{L}_\mathrm{retain}$ with $\mathcal{L}_\mathrm{forget}$ the loss on $\mathcal{D}_\mathrm{A}$ and $\mathcal{L}_\mathrm{retain}$ the loss on $\mathcal{D}_\mathrm{r}$.

## A.2 Computational Cost and Runtime of ReGUn and Baselines

In this section, we compare the compute structure and empirical runtime of the studied unlearning methods. The analysis considers one selected unlearning run after training, not an iterative stream of deletion requests where cached quantities could be amortized across requests.

**Computational Cost Analysis.** Let $T$ denote the number of unlearning steps, $n_\mathrm{r} = |\mathcal{D}_\mathrm{r}|$, $n_\mathrm{f} = |\mathcal{D}_\mathrm{f}|$, and $n_\mathrm{h} = |\mathcal{D}_\mathrm{h}|$. We write $b_\mathrm{r} = |B_\mathrm{r}|$, $b_\mathrm{f} = |B_\mathrm{f}|$, and $b_\mathrm{h} = |B_\mathrm{h}|$ for retain, forget, and held-out reference minibatches, with $b_\mathrm{h} = b_\mathrm{f}$ in the default REGUN implementation. Let $K$ be the number of classes and $P$ the number of model parameters. Finally, let $C_G(b)$ denote the cost of one gradient update on a batch of size $b$, including the forward pass, backward pass, loss computation, and optimizer update, and let $C_F(b)$ denote the cost of a no-gradient forward pass. Loss computations such as cross-entropy or KL terms are included in $C_G$ and are not counted separately.

Note that the number of unlearning steps $T$ should be interpreted together with the dataset that drives the update schedule. If a method is run for $E_\mathrm{u}$ unlearning epochs over a set of size $n_\mathrm{s}$ with batch size $b_\mathrm{s}$, then $T \approx E_\mathrm{u}\lceil n_\mathrm{s}/b_\mathrm{s}\rceil$. This matters because some methods naturally iterate over $\mathcal{D}_\mathrm{f}$, whereas retain-only fine-tuning may iterate over the much larger $\mathcal{D}_\mathrm{r}$. For comparability, in our evaluations all methods with explicit iterative updates use a forget-set-sized update budget, with retain minibatches sampled as needed.

FINETUNE, NEGGRAD, and NEGGRAD+ mainly differ in which data enter an otherwise standard "first-order" update: retain batches, forget batches, or both retain and forget batches, respectively. Methods such as $\ell_1$-SPARSE follow the same first-order optimization path but add parameter-wise regularization, with $b_\mathrm{upd}$ denoting the batch size used by the corresponding update objective, typically $b_\mathrm{upd} = b_\mathrm{r}$. Its runtime is therefore also mainly driven by the dense forward/backward computation, with the additional parameter-wise regularization remaining comparatively small. Their costs can be summarized as

$$\mathcal{T}_\mathrm{FINETUNE} \approx TC_G(b_\mathrm{r}), \ \mathcal{T}_\mathrm{NEGGRAD} \approx TC_G(b_\mathrm{f}), \ \mathcal{T}_\mathrm{NEGGRAD+} \approx TC_G(b_\mathrm{f} + b_\mathrm{r}), \ \mathcal{T}_{\ell_1\text{-SPARSE}} \approx TC_G(b_\mathrm{upd}) + \mathcal{O}(TP).$$

The remaining baselines introduce method-specific stages. SSD estimates parameter-importance statistics, typically Fisher-based over $\mathcal{D}_\mathrm{r} \cup \mathcal{D}_\mathrm{f}$ and $\mathcal{D}_\mathrm{f}$ in the default single-request setting, and then applies a comparatively cheap parameter-wise dampening step, of cost $\mathcal{O}(P)$. The dominant cost comes from the importance-estimation stage, which can require expensive per-sample gradients. Let $S \subseteq \mathcal{D}_\mathrm{train}$ denote the dataset for importance scores; if the gradients are computed in minibatches of size $b$, we write $\mathcal{T}_\mathrm{Fisher}(S) \approx \left\lceil \frac{|S|}{b} \right\rceil C_\nabla(b)$, where $C_\nabla(b)$ is the cost of computing these gradients, without optimizer updates. The final SSD cost is

$$\mathcal{T}_\mathrm{SSD} \approx \mathcal{T}_\mathrm{Fisher}(\mathcal{D}_\mathrm{r} \cup \mathcal{D}_\mathrm{f}) + \mathcal{T}_\mathrm{Fisher}(\mathcal{D}_\mathrm{f}) + \mathcal{O}(P).$$

SALUN similarly adds a preprocessing stage to construct a saliency mask before unlearning. The first term captures the one-time gradient-based mask-construction pass over forget data, while the second term is the subsequent retain–forget unlearning phase. Unless sparsity is exploited structurally, the mask changes which parameters are updated but does not substantially reduce the dense forward/backward cost of each update:

$$\mathcal{T}_\mathrm{SALUN} \approx \left\lceil \frac{n_\mathrm{f}}{b_\mathrm{f}} \right\rceil C_\nabla(b_\mathrm{f}) + TC_G(b_\mathrm{f} + b_\mathrm{r}).$$

AMUN adds a different source of overhead: adversarial-example generation for forget samples. Compared with standard first-order methods, the additional cost therefore comes primarily from constructing adversarial forget samples, not from the final unlearning update itself. Its adversarial cost $\mathcal{T}_\mathrm{adv}$ scales with the number of attack steps and the number of forget examples:

$$\mathcal{T}_\mathrm{AMUN} \approx \mathcal{T}_\mathrm{adv} + TC_G(b_\mathrm{upd}).$$

LUR, in contrast, introduces overhead through an inner-update or bilevel structure, which can make each unlearning step more expensive than a single standard first-order update. If $S_{\text{in}}$ inner updates are used per outer step, its cost is at least

$$\mathcal{T}_{\text{LUR}} \gtrsim T(S_{\text{in}} + 1)C_G(b_{\text{upd}}).$$

Finally, REGUN follows the same first-order retain–forget update structure as NEGGRAD+, but replaces the loss-ascent forget target with a held-out reference distribution. At each iteration, REGUN samples $B_{\text{f}}$ and $B_{\text{r}}$, constructs $q(B_{\text{f}})$ from a held-out batch $B_{\text{h}}$ using the reference model $f_\phi$, and performs one gradient update on the combined objective. Its cost over $T$ unlearning steps is

$$\mathcal{T}_{\text{REGUN}} = T\left[C_G(b_{\text{f}} + b_{\text{r}}) + C_F(b_{\text{h}}) + \mathcal{O}(b_{\text{h}}K)\right],$$

with $b_{\text{h}} = b_{\text{f}}$ by default. The first term is the standard gradient update through the current model, shared with retain–forget first-order methods such as NEGGRAD+. The second term is the no-gradient reference-model forward pass on the held-out batch. The final term only accounts for averaging the held-out probability vectors into the batch-level reference distribution, which is in practice negligible. Therefore, the main additional operation in REGUN relative to NEGGRAD+ is one no-gradient reference-model forward pass per unlearning step. If the reference predictions $p_\phi(\cdot \mid x)$ for $x \in \mathcal{D}_{\text{h}}$ are precomputed, this overhead reduces to sampling and averaging cached $K$-dimensional vectors:

$$\mathcal{T}_{\text{REGUN}}^{\text{cached}} = T\left[C_G(b_{\text{f}} + b_{\text{r}}) + \mathcal{O}(b_{\text{h}}K)\right],$$

with one-time preprocessing cost $C_F(n_{\text{h}})$ and storage $\mathcal{O}(n_{\text{h}}K)$. Overall, REGUN remains close to standard first-order retain–forget unlearning methods in leading-order complexity. Its main extra cost is reference construction from held-out data, whereas other advanced baselines introduce importance-estimation passes, saliency-mask construction, adversarial-example generation, or inner-loop computation.

**Empirical Runtime Results.** Figure 4 complements the theoretical cost analysis with empirical wall-clock runtimes for all studied methods. All measurements here, and in the full manuscript, were taken on a single NVIDIA GeForce RTX 4090 GPU. For iterative methods, one unlearning epoch corresponds to a forget-set-sized update budget; we use 10 such epochs for ResNet-18 and 20 for Swin-T, sampling retain minibatches as needed. We exclude all validation steps to isolate the unlearning phase cost. The results are indicative runtime comparisons rather than fully optimized method benchmarks, as implementations were not individually tuned for runtime.

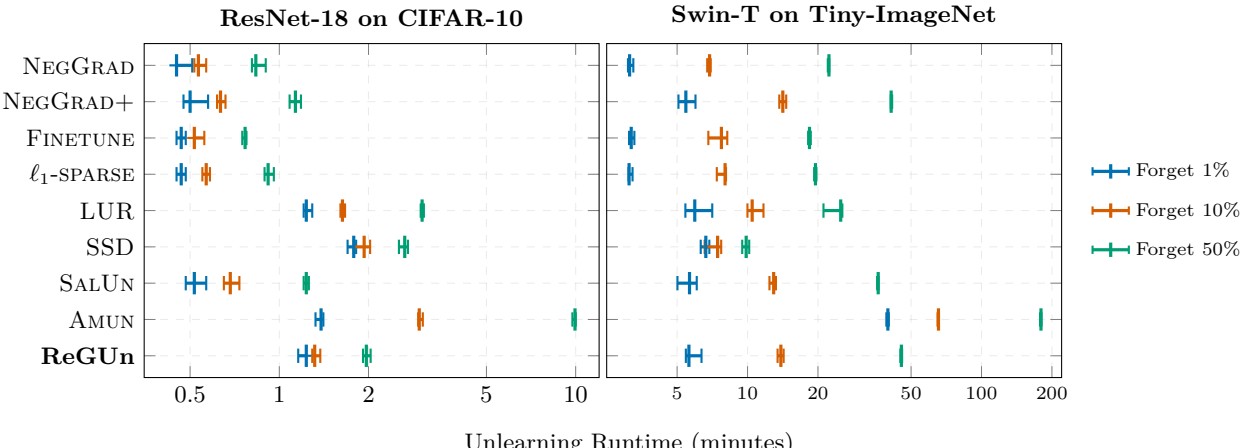

Figure 4: **Empirical Runtime Results.** Wall-clock runtime of the studied unlearning methods for the unlearning phase. Central vertical bars show the median over 15 runs and horizontal whiskers indicate the interquartile range (25th–75th percentile).

# B  Supplementary Experimental Results

## B.1  Ablation Details and Results

We provide full details and extended results for the ablation studies from Section 5. To make the ablations directly comparable, we use a fixed hyperparameter setting for all variants within each model–dataset and forget-fraction configuration, rather than conducting an independent hyperparameter search for every ablated variant. This protocol reduces confounding from variant-specific tuning and highlights the effect of the design choice under study. Consequently, the reported ablation numbers may differ slightly from the main benchmark results, where hyperparameters are selected for the full method comparison.

The fixed settings are as follows. For ResNet-18 on CIFAR-10, we use retain weight $w = 0.8$ and learning rate 0.005 for both 1% and 10% forgetting, and $w = 0.9$ with learning rate 0.05 for 50% forgetting. For Swin-T on Tiny-ImageNet, we use $w = 0.9$ and learning rate 0.0005 for 1% forgetting, and $w = 0.7$ with learning rate 0.0005 for both 10% and 50% forgetting.

Tables 5 and 6 report the complete teacher ablation results, with the same metric suite as the main results. We define the two non-default selector variants used in the selector ablation, $\textsc{RefDist}_{\text{uni}}$ in Algorithm 3 and $\textsc{RefDist}_{\text{1NN}}$ in Algorithm 4, and report their full metric breakdown in Tables 7 and 8. Finally, Tables 9 and 10 report the held-out-size ablation results, complementing the aggregate trends shown in Figure 3.

| | RA | FA | TA | $\text{RMIA}_{\text{AUC}}$ | $\text{Gap}_{\text{Avg}}$ | $\text{Gap}^{\text{F-U}}_{\text{Avg}}$ | $\text{SMIA}_{\text{AUC}}$ | $\text{Gap}^{\text{SMIA}}_{\text{Avg}}$ |
|---|---|---|---|---|---|---|---|---|
| **Forget 1%** | | | | | | | | |
| Oracle Retrained | 99.77 ±0.05 | 97.93 ±0.51 | 91.95 ±0.03 | 51.08 ±1.20 | 1.70 ±0.65 | 1.88 ±0.69 | **48.92** ±1.34 | 2.05 ±0.53 |
| Online Student | 99.63 ±0.13 | **97.19** ±0.78 | 91.83 ±0.15 | **50.53** ±1.15 | **1.47** ±0.67 | 1.72 ±0.35 | 48.77 ±0.96 | **1.97** ±0.50 |
| EMA 0.99 | 99.70 ±0.09 | 97.63 ±0.90 | 91.98 ±0.09 | 50.78 ±1.07 | 1.56 ±0.62 | 1.72 ±0.43 | 48.89 ±0.92 | 1.99 ±0.50 |
| EMA 0.999 | 99.70 ±0.09 | 97.70 ±0.84 | 91.99 ±0.07 | 50.77 ±1.08 | 1.58 ±0.64 | 1.71 ±0.43 | 48.89 ±0.93 | 2.01 ±0.50 |
| Frozen Base | **99.81** ±0.03 | 97.70 ±0.71 | **92.15** ±0.17 | 50.71 ±0.95 | 1.49 ±0.81 | **1.59** ±0.47 | 48.42 ±0.93 | 2.06 ±0.49 |
| **Forget 10%** | | | | | | | | |
| Oracle Retrained | 98.39 ±0.48 | 97.13 ±0.85 | 90.45 ±0.63 | **50.45** ±0.91 | 2.02 ±0.15 | 1.81 ±0.16 | **50.69** ±0.93 | 2.16 ±0.39 |
| Online Student | 97.65 ±1.56 | 96.30 ±2.01 | 89.83 ±1.72 | 50.95 ±0.73 | 2.28 ±0.28 | 2.37 ±0.65 | 51.23 ±0.75 | 2.43 ±0.80 |
| EMA 0.99 | 97.22 ±1.64 | 95.90 ±2.23 | 89.43 ±1.62 | 50.70 ±0.85 | 2.41 ±0.41 | 2.44 ±0.71 | 50.96 ±0.73 | 2.47 ±0.83 |
| EMA 0.999 | 96.69 ±2.30 | 95.05 ±3.22 | 88.77 ±2.32 | 50.78 ±0.85 | 2.85 ±1.06 | 2.81 ±1.03 | 51.19 ±0.91 | 2.61 ±1.18 |
| Frozen Base | **98.47** ±0.75 | 97.28 ±1.02 | **90.63** ±0.96 | 50.46 ±1.05 | **1.99** ±0.26 | **1.72** ±0.48 | 50.72 ±0.91 | **2.14** ±0.48 |
| **Forget 50%** | | | | | | | | |
| Oracle Retrained | **98.32** ±0.33 | 92.58 ±0.35 | **89.45** ±0.61 | 52.19 ±0.12 | **1.54** ±0.26 | **1.40** ±0.49 | 51.43 ±0.28 | **1.35** ±0.27 |
| Online Student | 97.76 ±0.64 | 92.03 ±1.01 | 88.62 ±0.60 | 52.13 ±0.33 | 1.74 ±0.19 | 1.79 ±0.36 | 51.50 ±0.24 | 1.58 ±0.38 |
| EMA 0.99 | 98.16 ±0.53 | 92.30 ±0.63 | 89.04 ±0.77 | 52.18 ±0.05 | 1.62 ±0.07 | 1.61 ±0.39 | 51.55 ±0.12 | 1.46 ±0.34 |
| EMA 0.999 | 97.76 ±0.34 | **91.68** ±0.78 | 88.39 ±0.65 | 52.17 ±0.55 | 1.72 ±0.10 | 1.93 ±0.38 | 51.62 ±0.52 | 1.58 ±0.35 |
| Frozen Base | 97.94 ±0.46 | 92.05 ±1.05 | 88.81 ±0.82 | **52.07** ±0.25 | 1.64 ±0.15 | 1.67 ±0.33 | **51.42** ±0.69 | 1.47 ±0.43 |

Table 5: **Teacher ablation full metrics for ResNet-18 on CIFAR-10. Bold** and underlined denote best and second best, where "best" denotes smallest gap to Retrain.

| | RA | FA | TA | RMIA$_{\text{AUC}}$ | Gap$_{\text{Avg}}$ | Gap$_{\text{Avg}}^{\text{F-U}}$ | SMIA$_{\text{AUC}}$ | Gap$_{\text{Avg}}^{\text{SMIA}}$ |
|---|---|---|---|---|---|---|---|---|
| **Forget 1%** | | | | | | | | |
| Oracle Retrained | **98.05** ±0.21 | 63.59 ±4.28 | 51.24 ±1.03 | 57.24 ±1.67 | 5.76 ±0.42 | 8.54 ±0.47 | **48.38** ±0.52 | 3.96 ±1.16 |
| Online Student | 97.77 ±0.45 | 62.00 ±1.74 | 51.23 ±0.88 | 56.15 ±0.28 | 4.75 ±0.33 | 8.00 ±0.81 | 47.82 ±0.20 | 3.77 ±0.62 |
| EMA 0.99 | 97.93 ±0.16 | 62.67 ±2.80 | 51.61 ±0.39 | 56.89 ±0.40 | 5.06 ±0.04 | 8.19 ±0.63 | 48.01 ±0.35 | 3.76 ±0.80 |
| EMA 0.999 | 97.89 ±0.19 | 62.78 ±1.85 | **51.71** ±0.34 | 56.78 ±0.46 | 4.90 ±0.07 | 8.08 ±0.70 | 47.96 ±0.28 | 3.78 ±0.59 |
| Frozen Base | 97.91 ±0.15 | **61.56** ±0.29 | 51.23 ±0.23 | **56.07** ±0.84 | 4.68 ±0.47 | **7.97** ±0.65 | 48.21 ±0.04 | **3.53** ±0.37 |
| **Forget 10%** | | | | | | | | |
| Oracle Retrained | 98.93 ±0.11 | 63.02 ±0.96 | 52.06 ±0.89 | **50.04** ±0.77 | **3.00** ±0.22 | 3.73 ±0.31 | 45.85 ±0.76 | 4.06 ±0.41 |
| Online Student | 98.84 ±0.11 | 63.57 ±0.62 | 52.43 ±0.24 | 51.01 ±1.10 | 3.23 ±0.61 | 3.87 ±0.68 | **46.27** ±0.42 | 4.03 ±0.25 |
| EMA 0.99 | 98.87 ±0.09 | 62.97 ±0.58 | 52.41 ±0.33 | 50.82 ±1.02 | 3.08 ±0.58 | 3.89 ±0.60 | 46.01 ±0.40 | **3.94** ±0.25 |
| EMA 0.999 | 98.90 ±0.06 | 63.31 ±0.34 | 52.45 ±0.23 | 50.79 ±1.09 | 3.10 ±0.48 | 3.76 ±0.59 | 45.95 ±0.46 | 4.02 ±0.22 |
| Frozen Base | **98.95** ±0.12 | 63.02 ±1.02 | **52.66** ±0.20 | 49.88 ±0.81 | **3.00** ±0.21 | 3.74 ±0.50 | 45.56 ±0.25 | 3.98 ±0.31 |
| **Forget 50%** | | | | | | | | |
| Oracle Retrained | **99.98** ±0.01 | 48.93 ±0.22 | **45.82** ±0.33 | 47.59 ±0.45 | 1.36 ±0.09 | 2.42 ±0.22 | 47.95 ±0.34 | 1.26 ±0.14 |
| Online Student | 99.97 ±0.01 | 47.03 ±0.20 | 44.96 ±0.59 | 47.49 ±0.77 | 1.79 ±0.08 | 2.90 ±0.23 | 47.74 ±0.56 | 1.71 ±0.22 |
| EMA 0.99 | **99.97** ±0.00 | 47.40 ±0.53 | 45.29 ±0.51 | 47.77 ±0.33 | 1.54 ±0.23 | 2.60 ±0.11 | 47.75 ±0.28 | 1.53 ±0.21 |
| EMA 0.999 | 99.97 ±0.01 | 47.24 ±0.42 | 45.22 ±0.48 | **48.03** ±0.32 | 1.53 ±0.12 | 2.50 ±0.06 | 47.92 ±0.37 | 1.54 ±0.19 |
| Frozen Base | 99.97 ±0.01 | **48.28** ±0.62 | 45.64 ±0.40 | 47.86 ±0.18 | **1.34** ±0.13 | 2.38 ±0.20 | **48.11** ±0.39 | **1.13** ±0.22 |

Table 6: **Teacher ablation full metrics for Swin-T on Tiny-ImageNet. Bold** and underlined denote best and second best, where "best" denotes smallest gap to RETRAIN.

| | RA | FA | TA | RMIA$_{\text{AUC}}$ | Gap$_{\text{Avg}}$ | Gap$_{\text{Avg}}^{\text{F-U}}$ | SMIA$_{\text{AUC}}$ | Gap$_{\text{Avg}}^{\text{SMIA}}$ |
|---|---|---|---|---|---|---|---|---|
| **Forget 1%** | | | | | | | | |
| Uniform | 99.70 ±0.08 | **97.48** ±0.90 | 91.97 ±0.06 | 50.74 ±0.93 | 1.52 ±0.31 | 1.70 ±0.25 | 48.60 ±1.06 | **2.03** ±0.52 |
| Feature-1NN | 99.71 ±0.08 | 97.56 ±0.59 | 91.95 ±0.15 | 50.80 ±0.92 | 1.55 ±0.50 | 1.74 ±0.27 | **48.63** ±1.16 | 2.04 ±0.51 |
| Class-Conditioned | **99.81** ±0.03 | 97.70 ±0.71 | **92.15** ±0.17 | **50.71** ±0.95 | **1.49** ±0.81 | **1.59** ±0.47 | 48.42 ±0.93 | 2.06 ±0.49 |
| **Forget 10%** | | | | | | | | |
| Uniform | 96.51 ±3.19 | **95.00** ±3.79 | 88.93 ±3.30 | **50.44** ±0.69 | 2.90 ±1.57 | 2.57 ±1.46 | 50.74 ±0.64 | 2.49 ±1.50 |
| Feature-1NN | 97.85 ±1.25 | 96.56 ±1.51 | 89.99 ±1.47 | **50.44** ±0.68 | 2.13 ±0.27 | 2.04 ±0.54 | 50.74 ±0.80 | 2.28 ±0.66 |
| Class-Conditioned | **98.47** ±0.75 | 97.28 ±1.02 | **90.63** ±0.96 | 50.46 ±1.05 | **1.99** ±0.26 | **1.72** ±0.48 | **50.72** ±0.91 | **2.14** ±0.48 |
| **Forget 50%** | | | | | | | | |
| Uniform | 97.66 ±1.03 | **91.76** ±1.14 | 88.62 ±1.01 | **51.91** ±0.38 | 1.64 ±0.22 | 1.68 ±0.56 | 51.13 ±0.26 | **1.45** ±0.50 |
| Feature-1NN | 97.87 ±0.26 | 91.85 ±0.20 | 88.63 ±0.14 | 51.92 ±0.16 | **1.61** ±0.12 | 1.68 ±0.08 | 51.30 ±0.06 | 1.46 ±0.20 |
| Class-Conditioned | **97.94** ±0.46 | 92.05 ±1.05 | **88.81** ±0.82 | 52.07 ±0.25 | 1.64 ±0.15 | 1.67 ±0.33 | 51.42 ±0.69 | 1.47 ±0.43 |

Table 7: **Selector ablation full metrics for ResNet-18 on CIFAR-10. Bold** and underlined denote best and second best, where "best" denotes smallest gap to RETRAIN.

| | RA | FA | TA | $\text{RMIA}_{\text{AUC}}$ | $\text{Gap}_{\text{Avg}}$ | $\text{Gap}_{\text{Avg}}^{\text{F-U}}$ | $\text{SMIA}_{\text{AUC}}$ | $\text{Gap}_{\text{Avg}}^{\text{SMIA}}$ |
|---|---|---|---|---|---|---|---|---|
| | | | | **Forget 1%** | | | | |
| Uniform | 97.90 ±0.30 | 60.63 ±2.78 | 50.83 ±0.48 | **55.77** ±0.61 | 4.88 ±0.15 | 8.01 ±0.70 | 47.51 ±0.46 | 4.00 ±0.80 |
| Feature-1NN | 97.74 ±0.06 | 61.00 ±3.00 | 51.07 ±0.81 | 56.56 ±2.06 | 5.35 ±0.29 | 8.29 ±0.22 | 48.15 ±1.12 | 3.73 ±0.90 |
| Class-Conditioned | **97.91** ±0.15 | **61.56** ±0.29 | **51.23** ±0.23 | 56.07 ±0.84 | **4.68** ±0.47 | **7.97** ±0.65 | **48.21** ±0.04 | **3.53** ±0.37 |
| | | | | **Forget 10%** | | | | |
| Uniform | 98.94 ±0.10 | **62.82** ±1.47 | 52.36 ±0.54 | **50.50** ±0.73 | 3.06 ±0.35 | 3.96 ±0.23 | 45.95 ±0.24 | **3.91** ±0.43 |
| Feature-1NN | **99.01** ±0.12 | 64.27 ±1.49 | **52.72** ±0.83 | 50.95 ±0.83 | 3.33 ±0.35 | 3.80 ±0.35 | 46.17 ±0.15 | 4.11 ±0.46 |
| Class-Conditioned | 98.95 ±0.12 | 63.02 ±1.02 | 52.66 ±0.20 | 49.88 ±0.81 | **3.00** ±0.21 | 3.74 ±0.50 | 45.56 ±0.25 | 3.98 ±0.31 |
| | | | | **Forget 50%** | | | | |
| Uniform | **99.97** ±0.01 | 47.02 ±1.02 | 44.88 ±0.77 | 47.42 ±0.18 | 1.82 ±0.41 | 2.98 ±0.26 | 47.86 ±0.24 | 1.70 ±0.33 |
| Feature-1NN | **99.97** ±0.01 | 49.50 ±0.52 | 45.72 ±0.50 | **48.27** ±0.36 | 1.36 ±0.18 | 2.13 ±0.52 | 48.35 ±0.25 | 1.33 ±0.20 |
| Class-Conditioned | 99.97 ±0.01 | **48.28** ±0.62 | 45.64 ±0.40 | 47.86 ±0.18 | **1.34** ±0.13 | 2.38 ±0.20 | 48.11 ±0.39 | **1.13** ±0.22 |

Table 8: **Selector ablation full metrics for Swin-T on Tiny-ImageNet. Bold** and underlined denote best and second best, where "best" denotes smallest gap to RETRAIN.

| | RA | FA | TA | $\text{RMIA}_{\text{AUC}}$ | $\text{Gap}_{\text{Avg}}$ | $\text{Gap}_{\text{Avg}}^{\text{F-U}}$ | $\text{SMIA}_{\text{AUC}}$ | $\text{Gap}_{\text{Avg}}^{\text{SMIA}}$ |
|---|---|---|---|---|---|---|---|---|
| | | | | **Forget 1%** | | | | |
| Held-out 0.1% | 99.54 ±0.16 | 95.78 ±1.39 | 91.73 ±0.14 | **47.60** ±1.40 | **1.73** ±0.42 | 2.24 ±0.77 | **46.30** ±0.81 | 2.28 ±0.56 |
| Held-out 0.3% | 99.55 ±0.08 | 94.74 ±1.99 | **91.83** ±0.09 | 47.46 ±1.59 | 1.78 ±0.76 | 2.26 ±0.78 | 45.72 ±1.27 | 2.14 ±0.71 |
| Held-out 1% | 99.57 ±0.04 | 95.19 ±1.81 | 91.77 ±0.13 | 47.15 ±1.68 | 1.87 ±0.71 | 2.45 ±0.91 | 45.46 ±1.37 | 2.32 ±0.69 |
| Held-out 3% | 99.58 ±0.04 | 95.11 ±1.74 | 91.77 ±0.17 | 47.17 ±1.69 | 1.84 ±0.72 | 2.44 ±0.94 | 45.47 ±1.36 | 2.30 ±0.68 |
| Held-out 10% | 99.58 ±0.04 | 95.11 ±1.94 | 91.76 ±0.16 | 47.15 ±1.67 | 1.88 ±0.74 | 2.45 ±0.91 | 45.46 ±1.37 | 2.30 ±0.71 |
| Held-out 31% | 99.58 ±0.04 | 95.11 ±1.94 | 91.74 ±0.18 | 47.15 ±1.70 | 1.89 ±0.76 | 2.46 ±0.94 | 45.47 ±1.38 | 2.31 ±0.71 |
| Held-out 100% | **99.58** ±0.04 | 95.19 ±1.81 | 91.78 ±0.14 | 47.15 ±1.68 | 1.86 ±0.71 | 2.44 ±0.91 | 45.45 ±1.37 | 2.32 ±0.69 |
| | | | | **Forget 10%** | | | | |
| Held-out 0.1% | 98.60 ±1.15 | **96.27** ±1.70 | 89.85 ±1.51 | 52.71 ±0.66 | 2.47 ±0.59 | 3.24 ±1.37 | 51.55 ±0.59 | 2.26 ±0.67 |
| Held-out 0.3% | 98.49 ±1.30 | 96.88 ±2.33 | 90.93 ±1.61 | **51.56** ±0.69 | 2.09 ±0.34 | 2.12 ±1.06 | 50.69 ±0.28 | 1.96 ±0.79 |
| Held-out 1% | 98.56 ±0.99 | 97.13 ±1.72 | 90.98 ±1.31 | 51.61 ±0.90 | 2.13 ±0.19 | 2.12 ±0.62 | 50.76 ±0.63 | 2.01 ±0.63 |
| Held-out 3% | 98.76 ±0.73 | 97.41 ±1.45 | 91.05 ±1.14 | 51.60 ±0.79 | 2.13 ±0.20 | 2.08 ±0.64 | 50.71 ±0.45 | 2.00 ±0.52 |
| Held-out 10% | 98.83 ±0.63 | 97.44 ±1.27 | **91.22** ±0.98 | 51.59 ±0.81 | **2.08** ±0.17 | 1.99 ±0.56 | **50.67** ±0.47 | **1.93** ±0.46 |
| Held-out 31% | 98.77 ±0.68 | 97.39 ±1.34 | 91.19 ±0.90 | 51.62 ±0.78 | 2.10 ±0.15 | 2.02 ±0.52 | 50.72 ±0.43 | 1.96 ±0.47 |
| Held-out 100% | **98.85** ±0.61 | 97.50 ±1.21 | 91.15 ±0.93 | 51.65 ±0.77 | 2.12 ±0.21 | 2.05 ±0.61 | 50.71 ±0.38 | 1.97 ±0.44 |
| | | | | **Forget 50%** | | | | |
| Held-out 0.1% | 97.97 ±0.87 | 92.50 ±1.08 | 89.20 ±0.78 | 52.31 ±0.21 | 1.71 ±0.21 | 1.59 ±0.39 | 51.38 ±0.12 | 1.47 ±0.44 |
| Held-out 0.3% | 98.39 ±0.69 | 92.88 ±0.81 | 89.87 ±0.68 | 52.34 ±0.58 | 1.61 ±0.06 | 1.42 ±0.12 | 51.49 ±0.36 | **1.32** ±0.37 |
| Held-out 1% | 98.52 ±0.58 | 92.90 ±0.61 | 89.55 ±0.23 | **51.97** ±0.48 | **1.50** ±0.14 | **1.25** ±0.04 | **51.30** ±0.17 | 1.33 ±0.29 |
| Held-out 3% | **98.72** ±0.32 | 93.38 ±0.58 | 89.87 ±0.60 | 52.33 ±0.31 | 1.57 ±0.12 | 1.26 ±0.19 | 51.34 ±0.21 | 1.33 ±0.29 |
| Held-out 10% | 98.10 ±0.38 | 92.12 ±0.60 | 89.03 ±0.52 | 52.07 ±0.31 | 1.56 ±0.22 | 1.56 ±0.20 | 51.53 ±0.76 | 1.42 ±0.34 |
| Held-out 31% | 98.27 ±0.41 | 92.62 ±0.57 | 89.37 ±0.41 | 52.53 ±0.05 | 1.67 ±0.15 | 1.61 ±0.34 | 51.56 ±0.11 | 1.43 ±0.27 |
| Held-out 100% | 97.57 ±0.85 | **91.55** ±1.44 | 88.27 ±1.17 | 52.19 ±0.23 | 1.77 ±0.30 | 1.99 ±0.60 | 51.55 ±0.46 | 1.61 ±0.55 |

Table 9: **Held-out subsample ablation full metrics for ResNet-18 on CIFAR-10. Bold** and underlined denote best and second best, where "best" denotes smallest gap to RETRAIN.

| | RA | FA | TA | $\text{RMIA}_{\text{AUC}}$ | $\text{Gap}_{\text{Avg}}$ | $\text{Gap}_{\text{Avg}}^{\text{F-U}}$ | $\text{SMIA}_{\text{AUC}}$ | $\text{Gap}_{\text{Avg}}^{\text{SMIA}}$ |
|---|---|---|---|---|---|---|---|---|
| | | | | **Forget 1%** | | | | |
| Held-out 0.1% | 97.78 ±0.15 | 54.41 ±2.79 | 50.44 ±0.57 | 54.85 ±1.31 | 6.18 ±0.91 | 7.75 ±0.58 | 46.97 ±0.65 | 5.82 ±0.81 |
| Held-out 0.3% | 97.79 ±0.47 | 57.26 ±1.22 | 51.03 ±0.96 | **54.76** ±1.83 | 5.30 ±0.63 | **7.41** ±1.19 | 47.00 ±1.03 | 4.95 ±0.60 |
| Held-out 1% | 97.98 ±0.25 | 60.19 ±1.70 | 50.91 ±0.98 | 55.86 ±1.59 | 4.97 ±0.75 | 8.02 ±1.21 | 47.75 ±0.99 | 4.01 ±0.66 |
| Held-out 3% | 97.91 ±0.31 | 60.44 ±0.68 | 51.02 ±0.70 | 56.01 ±1.16 | **4.79** ±0.65 | 8.04 ±1.12 | 47.96 ±0.79 | 3.88 ±0.48 |
| Held-out 10% | **98.04** ±0.28 | 61.59 ±2.02 | **51.24** ±1.05 | 56.47 ±1.09 | 4.86 ±0.56 | 8.16 ±1.01 | 48.12 ±0.54 | **3.52** ±0.69 |
| Held-out 31% | 98.00 ±0.22 | **61.59** ±0.86 | 51.10 ±0.82 | 56.40 ±1.61 | 4.87 ±0.47 | 8.20 ±1.07 | **48.14** ±0.76 | 3.56 ±0.50 |
| Held-out 100% | 97.99 ±0.33 | 61.30 ±1.98 | 51.08 ±0.77 | 56.43 ±1.19 | 5.03 ±0.19 | 8.22 ±0.75 | 48.06 ±0.65 | 3.61 ±0.66 |
| | | | | **Forget 10%** | | | | |
| Held-out 0.1% | 98.64 ±0.22 | 51.71 ±6.35 | 50.47 ±0.77 | 49.82 ±0.38 | 4.61 ±1.89 | 4.64 ±0.65 | 45.31 ±0.31 | 5.76 ±1.61 |
| Held-out 0.3% | 98.66 ±0.20 | 57.17 ±1.34 | 51.74 ±0.50 | 49.96 ±0.39 | 2.93 ±0.36 | 4.00 ±0.19 | 45.29 ±0.17 | 4.07 ±0.39 |
| Held-out 1% | 98.72 ±0.15 | **59.80** ±0.62 | 52.08 ±0.09 | 49.79 ±0.59 | **2.36** ±0.10 | 3.85 ±0.07 | 45.42 ±0.56 | **3.41** ±0.26 |
| Held-out 3% | 98.80 ±0.14 | 62.40 ±0.89 | 52.25 ±0.15 | 49.89 ±0.73 | 2.89 ±0.15 | 3.75 ±0.08 | 45.69 ±0.37 | 3.93 ±0.29 |
| Held-out 10% | 98.85 ±0.20 | 62.96 ±1.11 | 52.65 ±0.23 | 49.76 ±0.86 | 2.96 ±0.09 | 3.65 ±0.21 | 45.61 ±0.42 | 3.98 ±0.34 |
| Held-out 31% | **98.86** ±0.27 | 63.24 ±0.65 | **52.66** ±0.06 | **50.05** ±0.93 | 3.03 ±0.12 | 3.64 ±0.24 | **45.74** ±0.25 | 4.01 ±0.24 |
| Held-out 100% | 98.81 ±0.25 | 63.31 ±0.93 | 52.60 ±0.16 | 49.84 ±0.58 | 3.02 ±0.27 | **3.57** ±0.26 | 45.67 ±0.37 | 4.08 ±0.30 |
| | | | | **Forget 50%** | | | | |
| Held-out 0.1% | 99.98 ±0.01 | 33.59 ±1.00 | 39.70 ±0.41 | 52.77 ±0.41 | 6.37 ±0.36 | 5.37 ±0.37 | 49.97 ±0.27 | 5.82 ±0.28 |
| Held-out 0.3% | 99.98 ±0.01 | 43.52 ±2.08 | 43.20 ±0.82 | 52.48 ±0.56 | 2.94 ±0.88 | 3.47 ±0.68 | 49.98 ±0.38 | 2.46 ±0.57 |
| Held-out 1% | 99.96 ±0.01 | **48.76** ±0.88 | 44.95 ±0.60 | 52.01 ±0.21 | **1.37** ±0.11 | 2.36 ±0.44 | 49.87 ±0.33 | **0.96** ±0.28 |
| Held-out 3% | **99.98** ±0.01 | 50.67 ±0.50 | 45.55 ±0.41 | 51.62 ±0.41 | 1.52 ±0.16 | 1.86 ±0.47 | 49.87 ±0.51 | 1.28 ±0.21 |
| Held-out 10% | 99.97 ±0.02 | 50.86 ±0.22 | 45.65 ±0.49 | 51.69 ±0.05 | 1.56 ±0.08 | 1.84 ±0.33 | **50.05** ±0.24 | 1.26 ±0.16 |
| Held-out 31% | 99.97 ±0.00 | 51.08 ±0.42 | **46.04** ±0.40 | 51.50 ±0.39 | 1.47 ±0.09 | **1.56** ±0.28 | 49.94 ±0.27 | 1.24 ±0.17 |
| Held-out 100% | 99.97 ±0.02 | 50.86 ±0.51 | 45.71 ±0.38 | **51.21** ±0.32 | 1.42 ±0.16 | 1.58 ±0.41 | 49.95 ±0.27 | 1.27 ±0.18 |

Table 10: **Held-out subsample ablation full metrics for Swin-T on Tiny-ImageNet. Bold** and underlined denote best and second best, where "best" denotes smallest gap to RETRAIN.

---

**Algorithm 3** $\text{REFDIST}_{\text{uni}}$: Uniform Held-out Reference Distribution

1: **Input:** forget batch $B_{\text{f}}$, held-out set $\mathcal{D}_{\text{h}}$, reference model $f_\phi$
2: Sample $B_{\text{h}}$: draw $|B_{\text{f}}|$ examples uniformly from $\mathcal{D}_{\text{h}}$ (with replacement if needed)
3: Aggregate reference predictions: $q \leftarrow \frac{1}{|B_{\text{h}}|} \sum_{(x,\cdot) \in B_{\text{h}}} p_\phi(\cdot \mid x)$
4: **Output:** $q \in \Delta^K$

---

**Algorithm 4** $\text{REFDIST}_{\text{1NN}}$: Feature-Matched Held-out Reference Distribution

1: **Input:** forget batch $B_{\text{f}}$, held-out set $\mathcal{D}_{\text{h}}$, reference model $f_\phi$, feature map $g_\phi : \mathcal{X} \to \mathbb{R}^d$ induced by $f_\phi$ for nearest-neighbor retrieval
2: Compute normalized held-out features $\bar{g}_\phi(x) = g_\phi(x)/\|g_\phi(x)\|_2$ for all $(x,\cdot) \in \mathcal{D}_{\text{h}}$
3: Initialize $B_{\text{h}} \leftarrow \emptyset$
4: **for** each $(x_i,\cdot) \in B_{\text{f}}$ **do**
5:     Compute normalized forget feature $\bar{g}_\phi(x_i) = g_\phi(x_i)/\|g_\phi(x_i)\|_2$
6:     Find nearest held-out example: $j_i^\star \leftarrow \arg\max_{j:(x_j,\cdot) \in \mathcal{D}_{\text{h}}} \bar{g}_\phi(x_i)^\top \bar{g}_\phi(x_j)$
7:     Add $(x_{j_i^\star},\cdot)$ to $B_{\text{h}}$          ▷ keeping multiplicities
8: **end for**
9: Aggregate reference predictions: $q \leftarrow \frac{1}{|B_{\text{h}}|} \sum_{(x,\cdot) \in B_{\text{h}}} p_\phi(\cdot \mid x)$
10: **Output:** $q \in \Delta^K$

### B.2 Additional ResNet-18 Results on CIFAR-100 and Tiny-ImageNet

This section provides additional results for ResNet-18 on CIFAR-100 and Tiny-ImageNet using the same experimental protocol described in Section 4 and Appendix A.1. These experiments complement the main-text results by evaluating REGUN on additional ResNet-18 settings with larger label spaces and more challenging image data. Tables 11 and 12 report the complete results across all forget fractions, and Figure 5 shows the corresponding forgetting–utility trade-offs under 10% random forgetting.

The additional ResNet-18 results are in line with the main findings. Across CIFAR-100 and Tiny-ImageNet, REGUN is consistently among the strongest methods and achieves the best aggregate match to RETRAIN in most settings. The remaining cases follow the same pattern as in the main experiments: the best individual method can vary by dataset and forget fraction, but REGUN remains close to the strongest result while maintaining a stable balance between utility and membership indistinguishability. In particular, methods that preserve high test accuracy often retain a larger RMIA signal, whereas more aggressive forgetting methods can reduce this signal at the cost of utility. The trade-off curves in Figure 5 show the same behavior, with REGUN staying close to the RETRAIN RMIA reference while remaining in a competitive utility region. Overall, these additional benchmarks support that reference-guided distillation provides a robust unlearning signal beyond the primary settings.

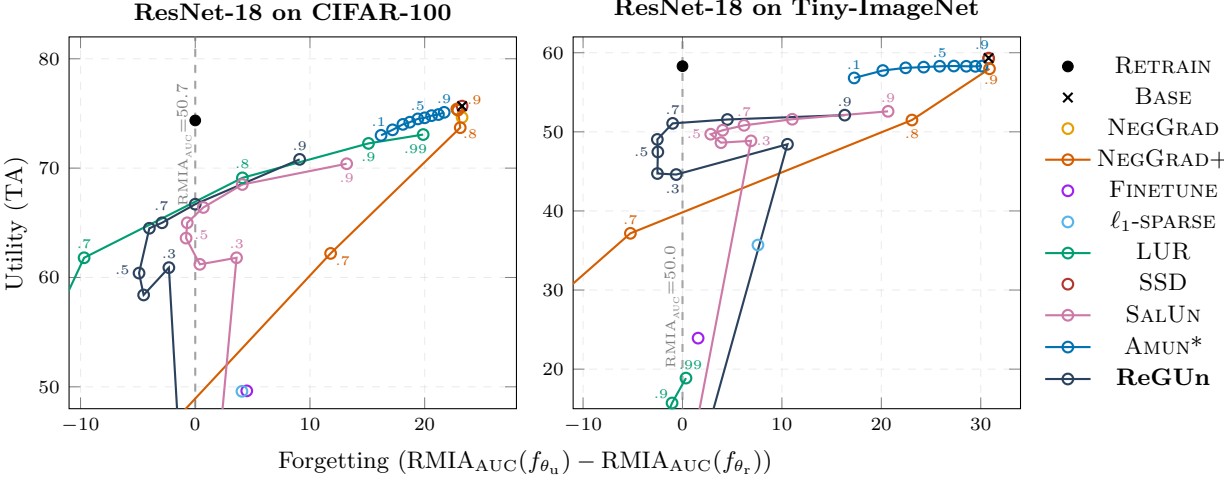

Figure 5: **Additional forgetting–utility trade-offs.** Trade-off curves for ResNet-18 on CIFAR-100 and Tiny-ImageNet under random 10% forgetting. The $y$-axis reports test accuracy and the $x$-axis reports the $\text{RMIA}_{\text{AUC}}$ gap to the retrain-from-scratch baseline; higher TA and proximity to the dashed vertical line is better. Where supported, we sweep $w \in [0.1, 0.9]$ in the retain–forget objective, with an additional $w = 0.99$ for LUR. Points are labeled by $w$; * marks variants reparameterized from the original loss to sweep $w$.

| | RA | FA | TA | $\text{RMIA}_{\text{AUC}}$ | $\text{Gap}_{\text{Avg}}$ | $\text{Gap}_{\text{Avg}}^{\text{F-U}}$ | $\text{SMIA}_{\text{AUC}}$ | $\text{Gap}_{\text{Avg}}^{\text{SMIA}}$ |
|---|---|---|---|---|---|---|---|---|
| | | | | **Forget 1%** | | | | |
| RETRAIN | 99.98 ±0.01 | 74.81 ±1.22 | 75.33 ±0.29 | 49.56 ±1.56 | 0.00 ±0.00 | 0.00 ±0.00 | 48.73 ±0.78 | 0.00 ±0.00 |
| BASE | 99.98 ±0.01 | 100.00 ±0.00 | 75.52 ±0.36 | 76.09 ±1.32 | 12.84 ±1.28 | 13.47 ±0.68 | 75.22 ±0.05 | 12.97 ±0.38 |
| NEGGRAD | 99.02 ±0.84 | 96.37 ±1.45 | 72.35 ±1.90 | 73.79 ±0.43 | 12.26 ±1.39 | 13.44 ±1.30 | 68.36 ±2.31 | 11.28 ±0.93 |
| NEGGRAD+ | 97.38 ±1.84 | 90.81 ±2.87 | 69.10 ±2.39 | 69.96 ±1.57 | 11.13 ±1.38 | 13.15 ±1.76 | 64.29 ±2.48 | 10.10 ±1.27 |
| FINETUNE | 91.98 ±3.58 | 75.63 ±4.69 | 66.67 ±1.27 | 59.76 ±3.26 | 7.04 ±1.15 | 9.27 ±1.75 | 54.46 ±2.58 | 5.80 ±1.68 |
| $\ell_1$-SPARSE | 92.44 ±1.01 | 73.56 ±3.27 | 66.30 ±0.52 | 59.78 ±0.85 | 7.02 ±0.29 | 9.46 ±1.06 | 53.91 ±1.48 | 5.75 ±1.01 |
| LUR | 77.23 ±0.84 | **74.44** ±2.04 | 61.41 ±0.66 | 56.72 ±0.85 | 11.25 ±0.38 | 10.38 ±0.48 | 55.27 ±1.24 | 10.90 ±0.75 |
| SSD | **99.61** ±0.46 | 98.89 ±0.89 | **75.12** ±0.48 | 75.78 ±1.32 | 12.63 ±1.22 | 13.23 ±1.13 | 74.09 ±1.19 | 12.51 ±0.55 |
| SALUN | 98.70 ±0.90 | 77.33 ±5.41 | 67.33 ±2.33 | **51.18** ±0.85 | 3.29 ±0.83 | **4.65** ±1.02 | 45.64 ±1.25 | 3.72 ±1.57 |
| AMUN | 96.09 ±3.71 | 76.74 ±10.38 | 70.43 ±4.79 | 57.91 ±4.97 | 5.96 ±1.15 | 6.46 ±1.44 | **51.65** ±6.55 | **3.41** ±3.44 |
| **ReGUn** | 99.53 ±0.03 | 76.89 ±0.97 | 69.23 ±0.60 | 46.91 ±0.98 | **3.01** ±0.28 | 4.73 ±0.87 | 41.39 ±0.77 | 3.99 ±0.50 |
| | | | | **Forget 10%** | | | | |
| RETRAIN | 99.98 ±0.00 | 75.59 ±0.27 | 74.36 ±0.01 | 50.71 ±0.22 | 0.00 ±0.00 | 0.00 ±0.00 | 50.32 ±0.43 | 0.00 ±0.00 |
| BASE | 99.98 ±0.00 | 99.99 ±0.01 | 75.67 ±0.17 | 73.97 ±0.42 | 12.24 ±0.20 | 12.28 ±0.27 | 75.41 ±0.38 | 12.70 ±0.16 |
| NEGGRAD | 99.86 ±0.12 | 99.74 ±0.12 | 74.55 ±0.77 | 74.35 ±0.84 | 12.11 ±0.21 | 12.08 ±0.28 | 73.83 ±0.60 | 11.99 ±0.28 |
| NEGGRAD+ | 99.86 ±0.11 | 98.79 ±0.65 | **74.23** ±0.87 | 73.72 ±0.51 | 11.74 ±0.15 | 11.81 ±0.39 | 72.09 ±0.80 | 11.31 ±0.36 |
| FINETUNE | 92.85 ±0.84 | 76.88 ±0.86 | 65.57 ±0.36 | 61.33 ±0.45 | 6.96 ±0.12 | 9.70 ±0.17 | 55.88 ±0.33 | 5.69 ±0.35 |
| $\ell_1$-SPARSE | 92.64 ±0.60 | 76.45 ±1.03 | 65.55 ±0.24 | 61.02 ±0.36 | 6.83 ±0.16 | 9.55 ±0.12 | 55.79 ±0.50 | 5.62 ±0.35 |
| LUR | 93.96 ±0.58 | **75.25** ±0.40 | 69.48 ±0.17 | 55.04 ±0.69 | **3.91** ±0.23 | 4.60 ±0.39 | 52.51 ±0.36 | **3.36** ±0.24 |
| SSD | **99.98** ±0.00 | 99.99 ±0.01 | 75.67 ±0.17 | 73.97 ±0.42 | 12.24 ±0.20 | 12.28 ±0.27 | 75.41 ±0.38 | 12.70 ±0.16 |
| SALUN | 98.55 ±1.42 | 89.81 ±3.97 | 67.74 ±2.40 | 55.26 ±0.83 | 6.71 ±0.19 | 5.59 ±1.56 | 52.97 ±0.94 | 6.23 ±1.24 |
| AMUN | 92.81 ±1.05 | 63.05 ±1.45 | 65.73 ±0.88 | **50.75** ±0.15 | 7.11 ±0.79 | **4.36** ±0.50 | 47.23 ±0.25 | 7.86 ±0.52 |
| **ReGUn** | 97.79 ±1.48 | 83.61 ±5.07 | 66.55 ±2.21 | 50.10 ±1.41 | 4.80 ±0.08 | 4.50 ±1.51 | **48.85** ±0.52 | 4.87 ±1.44 |
| | | | | **Forget 50%** | | | | |
| RETRAIN | 99.99 ±0.00 | 65.81 ±0.10 | 65.77 ±0.75 | 50.43 ±0.43 | 0.00 ±0.00 | 0.00 ±0.00 | 49.86 ±0.34 | 0.00 ±0.00 |
| BASE | 99.98 ±0.01 | 99.98 ±0.00 | 75.69 ±0.15 | 67.83 ±0.36 | 15.38 ±0.05 | 13.66 ±0.07 | 75.29 ±0.18 | 17.38 ±0.22 |
| NEGGRAD | 99.98 ±0.01 | 99.97 ±0.02 | 75.33 ±0.18 | 67.94 ±0.23 | 15.31 ±0.07 | 13.53 ±0.14 | 75.08 ±0.26 | 17.24 ±0.22 |
| NEGGRAD+ | 91.48 ±1.63 | 66.63 ±1.31 | 60.85 ±1.19 | 56.68 ±0.33 | 5.21 ±0.78 | 5.58 ±1.04 | 53.38 ±0.36 | 4.44 ±0.64 |
| FINETUNE | 92.29 ±0.63 | 70.71 ±0.83 | 61.64 ±0.59 | 59.43 ±0.56 | 6.43 ±0.48 | 6.56 ±0.84 | 55.09 ±0.19 | 5.49 ±0.37 |
| $\ell_1$-SPARSE | 92.46 ±1.48 | 69.83 ±2.19 | 62.60 ±0.87 | 58.05 ±1.46 | 5.58 ±0.57 | 5.39 ±0.89 | 54.32 ±0.73 | 4.80 ±0.75 |
| LUR | 95.04 ±0.80 | 66.88 ±3.28 | **62.75** ±1.64 | **54.45** ±2.24 | 3.71 ±0.16 | **3.52** ±0.36 | 52.35 ±1.17 | 2.88 ±1.00 |
| SSD | **99.98** ±0.01 | 99.98 ±0.00 | 75.68 ±0.15 | 67.83 ±0.36 | 15.37 ±0.05 | 13.65 ±0.07 | 75.29 ±0.18 | 17.38 ±0.22 |
| SALUN | 95.69 ±1.61 | 69.44 ±2.37 | 59.02 ±1.63 | 56.53 ±0.32 | 5.19 ±0.53 | 6.42 ±1.45 | 54.02 ±0.24 | 4.71 ±0.85 |
| AMUN | 92.16 ±0.97 | **65.74** ±0.44 | 62.45 ±0.81 | 55.05 ±0.55 | 4.04 ±0.46 | 3.97 ±0.51 | **51.51** ±0.49 | 3.22 ±0.41 |
| **ReGUn** | 97.91 ±0.72 | 68.84 ±1.62 | 61.55 ±0.51 | 55.16 ±0.83 | **3.51** ±0.48 | 4.47 ±0.99 | 51.93 ±0.47 | **2.85** ±0.52 |

Table 11: **Forgetting and utility results for ResNet-18 on CIFAR-100.** Studied under 1%, 10%, and 50% random forgetting. RA/FA/TA denote retain, forget, and test accuracy; $\text{RMIA}_{\text{AUC}}$ and $\text{SMIA}_{\text{AUC}}$ report membership inference performance, with 50% corresponding to chance-level discrimination. Gap metrics measure deviation from RETRAIN, so lower is better. Results are mean ± std over three seeds, with all values in percent. **Bold** and underlined denote best and second best, where "best" denotes smallest gap to RETRAIN.

| | RA | FA | TA | $\text{RMIA}_{\text{AUC}}$ | $\text{Gap}_{\text{Avg}}$ | $\text{Gap}_{\text{Avg}}^{\text{F-U}}$ | $\text{SMIA}_{\text{AUC}}$ | $\text{Gap}_{\text{Avg}}^{\text{SMIA}}$ |
|---|---|---|---|---|---|---|---|---|
| | | | | **Forget 1%** | | | | |
| Retrain | 99.98 ±0.00 | 58.52 ±0.71 | 59.54 ±0.21 | 49.28 ±1.14 | 0.00 ±0.00 | 0.00 ±0.00 | 49.84 ±0.85 | 0.00 ±0.00 |
| Base | 99.98 ±0.00 | 99.96 ±0.06 | 59.64 ±0.44 | 81.56 ±0.56 | 18.50 ±0.61 | 16.27 ±0.90 | 84.68 ±0.16 | 19.10 ±0.31 |
| NegGrad | **99.98** ±0.00 | 99.72 ±0.24 | **58.85** ±0.60 | 81.44 ±0.69 | 18.29 ±0.67 | 16.11 ±1.07 | 83.66 ±0.09 | 18.93 ±0.33 |
| NegGrad+ | 95.68 ±0.05 | 67.22 ±2.20 | 49.40 ±0.67 | 62.23 ±1.09 | 8.80 ±0.59 | 11.23 ±0.45 | 59.47 ±1.74 | 8.19 ±0.77 |
| Finetune | 89.92 ±6.35 | 61.67 ±8.33 | 51.55 ±0.69 | 63.06 ±7.81 | 9.13 ±1.00 | 10.57 ±3.13 | 57.23 ±5.13 | 7.15 ±2.93 |
| $\ell_1$-sparse | 88.23 ±4.10 | **59.70** ±1.66 | 51.87 ±0.16 | 60.21 ±3.75 | 7.90 ±0.39 | 9.30 ±1.92 | 55.48 ±1.68 | 6.56 ±1.22 |
| LUR | 3.94 ±0.16 | 3.59 ±0.28 | 3.77 ±0.18 | **49.14** ±0.32 | 51.91 ±0.38 | 28.33 ±0.29 | **49.25** ±0.69 | 51.83 ±0.34 |
| SSD | 99.39 ±0.01 | 98.89 ±0.31 | 58.72 ±0.10 | 80.69 ±0.60 | 18.08 ±0.50 | 15.80 ±0.78 | 83.87 ±0.13 | 18.95 ±0.30 |
| SalUn | 98.39 ±0.31 | 54.89 ±2.36 | 50.36 ±0.16 | 49.46 ±2.00 | 4.11 ±0.98 | 5.48 ±0.24 | 46.99 ±0.60 | 4.31 ±0.68 |
| Amun | 96.95 ±2.82 | 68.74 ±2.53 | 52.44 ±1.14 | 64.41 ±2.00 | 8.87 ±0.30 | 11.11 ±0.79 | 61.70 ±1.56 | 8.05 ±1.10 |
| **ReGUn** | 99.74 ±0.01 | 60.22 ±2.51 | 52.37 ±0.77 | 49.52 ±0.05 | **2.32** ±0.30 | **3.79** ±0.26 | 46.99 ±0.17 | **2.99** ±0.72 |
| | | | | **Forget 10%** | | | | |
| | RA | FA | TA | $\text{RMIA}_{\text{AUC}}$ | $\text{Gap}_{\text{Avg}}$ | $\text{Gap}_{\text{Avg}}^{\text{F-U}}$ | $\text{SMIA}_{\text{AUC}}$ | $\text{Gap}_{\text{Avg}}^{\text{SMIA}}$ |
| Retrain | 99.99 ±0.00 | 58.57 ±0.37 | 58.30 ±0.29 | 49.97 ±0.49 | 0.00 ±0.00 | 0.00 ±0.00 | 50.22 ±0.29 | 0.00 ±0.00 |
| Base | 99.98 ±0.00 | 99.99 ±0.01 | 59.33 ±0.39 | 80.76 ±0.55 | 18.31 ±0.10 | 15.91 ±0.21 | 84.77 ±0.13 | 19.25 ±0.17 |
| NegGrad | **99.98** ±0.00 | 99.98 ±0.01 | 59.53 ±0.08 | 80.49 ±0.17 | 18.32 ±0.18 | 16.02 ±0.22 | 84.43 ±0.01 | 19.22 ±0.14 |
| NegGrad+ | 96.16 ±0.00 | 79.31 ±0.00 | 50.96 ±0.00 | 72.14 ±0.00 | 13.60 ±0.00 | 14.92 ±0.00 | 67.21 ±0.00 | 12.23 ±0.14 |
| Finetune | 84.26 ±1.37 | 56.58 ±0.72 | 50.15 ±0.02 | 57.32 ±0.49 | 8.35 ±0.35 | 7.74 ±0.48 | 54.30 ±0.26 | 7.49 ±0.42 |
| $\ell_1$-sparse | 88.98 ±2.21 | **59.18** ±1.24 | 50.41 ±0.54 | 60.24 ±2.37 | 7.56 ±0.18 | 9.08 ±1.25 | 55.78 ±1.00 | 6.27 ±0.71 |
| LUR | 21.50 ±1.25 | 19.50 ±0.76 | 19.37 ±0.90 | **49.84** ±0.59 | 39.25 ±0.61 | 19.72 ±0.37 | **49.89** ±0.25 | 39.21 ±0.46 |
| SSD | 99.98 ±0.00 | 99.99 ±0.00 | **59.40** ±0.00 | 80.31 ±0.00 | 18.41 ±0.00 | 16.12 ±0.00 | 84.70 ±0.00 | 19.25 ±0.14 |
| SalUn | 99.43 ±0.00 | 83.10 ±0.00 | 51.55 ±0.00 | 60.61 ±0.00 | 10.70 ±0.00 | 8.86 ±0.00 | 58.89 ±0.00 | 10.13 ±0.14 |
| Amun | 96.90 ±3.32 | 71.54 ±3.13 | 52.53 ±1.43 | 66.67 ±2.74 | 9.63 ±0.50 | 11.23 ±0.79 | 63.58 ±2.10 | 8.80 ±1.31 |
| **ReGUn** | 98.69 ±0.88 | 67.28 ±0.97 | 51.13 ±0.85 | 48.60 ±2.01 | **4.51** ±0.20 | **4.12** ±0.23 | 49.55 ±1.57 | **4.46** ±0.57 |
| | | | | **Forget 50%** | | | | |
| | RA | FA | TA | $\text{RMIA}_{\text{AUC}}$ | $\text{Gap}_{\text{Avg}}$ | $\text{Gap}_{\text{Avg}}^{\text{F-U}}$ | $\text{SMIA}_{\text{AUC}}$ | $\text{Gap}_{\text{Avg}}^{\text{SMIA}}$ |
| Retrain | 99.99 ±0.00 | 49.35 ±0.11 | 49.33 ±0.38 | 50.14 ±0.33 | 0.00 ±0.00 | 0.00 ±0.00 | 50.03 ±0.20 | 0.00 ±0.00 |
| Base | 99.98 ±0.00 | 99.99 ±0.00 | 59.46 ±0.15 | 76.03 ±0.05 | 21.67 ±0.16 | 18.01 ±0.25 | 84.77 ±0.27 | 23.88 ±0.14 |
| NegGrad | 0.47 ±0.04 | 0.45 ±0.05 | 0.46 ±0.01 | **49.93** ±0.01 | 49.44 ±0.05 | 24.64 ±0.07 | 49.35 ±0.29 | 49.49 ±0.13 |
| NegGrad+ | 91.61 ±1.41 | **50.10** ±1.07 | 44.89 ±1.41 | 56.71 ±0.29 | 5.11 ±0.37 | 5.50 ±0.44 | 53.99 ±0.15 | **4.38** ±0.58 |
| Finetune | 92.68 ±1.75 | 55.26 ±1.50 | 46.09 ±0.95 | 61.05 ±0.62 | 6.82 ±0.07 | 7.06 ±0.26 | 56.78 ±0.27 | 5.80 ±0.64 |
| $\ell_1$-sparse | 88.03 ±0.91 | 50.64 ±0.66 | 44.74 ±0.54 | 57.27 ±0.11 | 6.24 ±0.34 | 5.86 ±0.55 | 54.41 ±0.26 | 5.56 ±0.34 |
| LUR | 52.18 ±1.62 | 33.52 ±0.70 | 33.45 ±0.72 | 50.56 ±0.39 | 19.99 ±0.70 | 8.15 ±0.21 | **50.35** ±0.22 | 19.96 ±0.49 |
| SSD | **99.99** ±0.00 | 99.99 ±0.00 | 59.39 ±0.04 | 76.05 ±0.07 | 21.59 ±0.05 | 17.88 ±0.11 | 84.71 ±0.36 | 23.85 ±0.14 |
| SalUn | 98.85 ±0.22 | 58.70 ±0.75 | 45.53 ±0.38 | 61.14 ±0.17 | 6.32 ±0.21 | 7.40 ±0.17 | 57.49 ±0.09 | 5.44 ±0.24 |
| Amun | 95.79 ±3.61 | 73.10 ±7.62 | **51.22** ±3.67 | 66.60 ±3.47 | 11.78 ±2.72 | 9.59 ±3.39 | 65.55 ±4.32 | 11.34 ±2.54 |
| **ReGUn** | 98.20 ±0.92 | 54.61 ±11.64 | 43.34 ±1.07 | 55.16 ±0.58 | **5.01** ±2.51 | **5.50** ±0.44 | 55.81 ±4.96 | 4.71 ±3.18 |

Table 12: **Forgetting and utility results for ResNet-18 on Tiny-ImageNet.** Studied under 1%, 10%, and 50% random forgetting. RA/FA/TA denote retain, forget, and test accuracy; $\text{RMIA}_{\text{AUC}}$ and $\text{SMIA}_{\text{AUC}}$ report membership inference performance, with 50% corresponding to chance-level discrimination. Gap metrics measure deviation from Retrain, so lower is better. Results are mean ± std over three seeds, with all values in percent. **Bold** and underlined denote best and second best, where "best" denotes smallest gap to Retrain.

