# OpenReview forum: "Reference-Guided Machine Unlearning"
_TMLR — Under review for TMLR_

### Review · Reviewer_zDGM · 2026-06-23

**Summary Of Contributions:**

The paper proposes ReGUn, an instance-level unlearning method that distills forget samples toward held-out non-member reference predictions instead of degrading them. It is empirically competitive, but its novelty over prior distillation-based unlearning is limited, and its reliance on labeled held-out data plus batch-level mixed targets raises conceptual concerns.

**Audience:**

Yes

**Audience Explanation:**

Machine unlearning is an active topic connected to privacy, deletion requests, and membership inference. The paper proposes a simple held-out-reference distillation approach and evaluates it across several vision settings. Even though the evidence has limitations, the idea is relevant to researchers working on approximate unlearning, privacy evaluation, and distillation-based methods.

**Broader Impact Concerns:**

The broader impact discussion should explicitly discuss governance of the held-out set $D_h$, especially if it contains sensitive data or is reused across multiple unlearning requests.

**Claims And Evidence:**

No

**Claims Explanation:**

The evidence is promising, but it does not fully support the submission’s broader claims. The results show that ReGUn is competitive on the evaluated random instance-forgetting benchmarks, but they do not convincingly establish that it is a generally stronger, more principled, or more reliable unlearning approach.

1. The paper omits key teacher/KD-style unlearning baselines, especially Chundawat et al. (2023). Since ReGUn is itself a distillation-based method with a held-out reference target, this is one of the most relevant comparisons.

2. The novelty is somewhat overstated. The paper acknowledges that indistinguishability has appeared before. But, the specific contribution is narrower: operationalize it using a disjoint held-out set to construct $q(B_f)$ for forget-sample distillation.

3. The evidence for the reference target is incomplete. ReGUn uses the frozen base model as the reference teacher, but this model was trained on $D_f$. Thus, $q(B_f)$ is not necessarily a clean proxy for retrain-only behavior.

4. ReGUn uses one batch-level $q(B_f)$ for all forget samples in a minibatch. If $B_f$ contains multiple or imbalanced classes, all samples are pushed toward the same mixed or one dominated target, which may be inappropriate for minority-class or atypical samples.

5. The experiments focus on uniformly random instance forgetting. They do not test concentrated/imbalanced/rare-class/subgroup-specific/semantically clustered forget requests, where the batch-level reference construction is more likely to fail. This is more challenging scenario for instance-level unlearning while maintaining utility.

6. The method assumes access to a labeled disjoint held-out set $D_h$. The paper does not clearly justify how this set is obtained in deployment, since it seems that $D_h$ must be reserved before training.

7. The paper lacks a simple behavioral-masking baseline, such as a frozen model whose outputs are suppressed only for known forget examples. Without this diagnostic, it is unclear whether the metrics distinguish true unlearning from surface-level output suppression.

8. The evaluation relies mostly on accuracy and MIA AUC metrics. These are useful output-level diagnostics, but they do not show removal of influence from parameters, representations, gradients, or neighborhoods around the forgotten samples.

**Requested Changes:**

Critical:

1. Add KD-style unlearning baselines such as Bad Teacher / Chundawat et al. (2023).

2. Evaluate non-random forget requests, such as one or more of imbalanced, rare-class, subgroup-specific, or semantically clustered forget sets.

3. Clarify the novelty claim by separating prior indistinguishability ideas from the specific held-out-reference construction.

4. Either justify the batch-level target $q(B_f)$ or test alternatives such as per-class reference targets.

5. Clarify the role and source of $D_h$, including how it could be reserved in real deployment and whether it can be reused across multiple unlearning requests.

Strengthening:

6. Add a behavioral-masking diagnostic baseline to check whether the metrics distinguish unlearning from output suppression on known forget examples.

7. Expand evaluation beyond accuracy and MIA AUC, e.g., representation, gradient, influence, or neighborhood-based diagnostics.

8. Discuss the limitation that the frozen teacher was trained on (D_f) and is therefore not a clean retrain-only reference.

---

### Review · Reviewer_Tb6X · 2026-07-01

**Summary Of Contributions:**

This paper studies approximate instance-level machine unlearning for vision classifiers. The main premise is that many existing unlearning methods operationalize forgetting through degradation-based objectives, such as loss maximization, random labeling, or other mechanisms that reduce performance on the forget set. The authors argue that such degradation is not equivalent to making the model behave as if the forget samples had never been observed. Instead, they propose to optimize for distributional indistinguishability between forget samples and truly unseen non-member data.

To instantiate this idea, the paper introduces Reference-Guided Unlearning (ReGUn). For a forget minibatch, ReGUn samples a class-histogram-matched minibatch from a disjoint held-out set, evaluates these held-out examples using a frozen reference model, and averages the resulting predictions into a class-conditioned reference distribution. The unlearned model is then trained to match this reference distribution on the forget examples, while a standard retain-set cross-entropy loss is used to preserve utility. The method is evaluated on several image-classification settings, including CIFAR-10, CIFAR-100, and Tiny-ImageNet, with both ResNet-18 and Swin-T backbones and multiple forget fractions. The paper compares against a range of approximate unlearning baselines and reports utility, forget accuracy, membership-inference metrics, and gap-to-retraining metrics.

The main strengths of the paper are its clear motivation, simple and practical method, reasonably broad empirical evaluation, and useful ablations on the reference teacher, held-out selector, and held-out set size. The distinction between degradation and indistinguishability is well articulated and aligns naturally with membership-inference-based evaluation.

The main weaknesses are that the technical novelty is moderate, since the method is essentially a class-conditioned held-out soft-target distillation objective; the method relies on access to a disjoint labeled held-out set from the same distribution; and the empirical gains, while generally positive, are not uniformly dominant across all metrics and settings. The paper also does not provide formal unlearning guarantees, so it should be positioned clearly as an empirical approximate unlearning method rather than a certified unlearning mechanism.

**Audience:**

Yes

**Audience Explanation:**

Machine unlearning is an active and practically important topic, especially given increasing interest in privacy, data deletion, post-deployment model editing, and membership-inference-based evaluation. TMLR’s audience includes researchers interested in privacy-preserving machine learning, model editing, trustworthy ML, empirical deep learning methodology, and vision model behavior. At least some of this audience would likely find the paper’s findings useful.

The paper’s main conceptual point—that unlearning should aim for non-member-like behavior rather than merely degraded forget-set performance—is relevant beyond the specific implementation. The proposed method is also simple enough to serve as a practical baseline or design principle for future approximate unlearning work. The empirical observation that class-conditioned held-out reference distributions can produce stable forgetting behavior is likely to be of interest to researchers working on approximate unlearning, even if they do not adopt the exact method.

The audience interest is somewhat limited by the scope of the paper. The experiments focus on supervised vision classification with random instance-level forgetting and access to labeled same-distribution held-out data. The findings may not immediately generalize to language models, generative models, structured deletion requests, or settings without clean auxiliary data. Nevertheless, within empirical machine unlearning for vision classifiers, the paper addresses a meaningful problem and presents findings that are likely to be useful.

**Broader Impact Concerns:**

The paper studies machine unlearning, which has positive potential relevance to privacy, data deletion, and post-deployment data governance. I do not see severe broader-impact concerns.

**Claims And Evidence:**

Yes

**Claims Explanation:**

Overall, the main empirical claims are supported by reasonably clear evidence. The paper claims that reference-guided distillation can improve the forgetting–utility trade-off compared with several degradation-based or regularization-based approximate unlearning methods. This claim is largely supported by the experimental results: the authors evaluate multiple datasets, architectures, and forget fractions, and compare ReGUn with a diverse set of baselines, including NegGrad, NegGrad+, Finetune, L1-sparse fine-tuning, LUR, SSD, SalUn, and Amun. The use of both utility metrics and membership-inference-based forgetting metrics is appropriate for the stated goal of indistinguishability.
﻿

The evidence is particularly convincing in the higher-resolution Swin-T/Tiny-ImageNet setting, where ReGUn shows strong aggregate performance relative to the retrain-from-scratch baseline. The ablation studies on teacher choice, held-out sample selection, and held-out set size also support the authors’ design choices and help explain why the method works.

﻿
However, ReGUn is not uniformly best across all individual metrics or all forget fractions. In several cases, other baselines are competitive on test accuracy, forget accuracy, or membership-inference AUC. The authors rely substantially on aggregate gap-to-retraining metrics, and the relative weighting of these metrics is not fully justified. Therefore, the evidence supports the more moderate claim that ReGUn often provides a favorable empirical trade-off, but it does not support a claim of uniform superiority.

﻿
The paper also does not provide formal guarantees of equivalence to retraining, and the authors acknowledge this. The unlearning claim is therefore empirical rather than certified. This is acceptable for the scope of the paper, but the framing should remain precise.

**Requested Changes:**

Discuss the held-out data assumption. Since ReGUn relies on a disjoint labeled held-out set with class coverage, the authors should clarify when this assumption is realistic and how the method may behave with small, shifted, unlabeled, or incomplete held-out data.

Add or discuss more realistic forgetting settings. The main experiments use random instance-level forgetting. The paper would be stronger with structured deletion requests, such as class-conditional, clustered, hard-example, or user-level forgetting.

Justify the aggregate metrics more clearly. GapAvg is useful, but it gives implicit equal weight to utility, forget accuracy, and membership-inference behavior. The authors should explain this choice and provide more disaggregated interpretation.

---

### Review · Reviewer_KuCw · 2026-07-20

**Summary Of Contributions:**

This paper introduces ReGUN (Reference-Guided Unlearning), a framework for approximate machine unlearning in vision models. The key idea is to align the model's predictions on forget samples with a reference distribution derived from held-out data, rather than using degradation-based objectives like loss maximization. The authors argue that unlearning should optimize for indistinguishability from unseen data rather than simply making the model "wrong" on forget examples. The method distills forget-set predictions toward class-conditioned distributions constructed from disjoint held-out data, while preserving retain-set utility through cross-entropy supervision.

Strengths:

1.	Clear motivation and principle: The paper makes a compelling argument that approximate unlearning should prioritize distributional indistinguishability rather than performance degradation. This reframing is well-motivated and addresses a genuine gap in the literature.

2.	Comprehensive experimental evaluation: The paper evaluates across multiple datasets, architectures, and forget fractions. The comparison against diverse baselines is thorough.

3.	The paper systematically stress-tests its own design choices: the reference-model source, the held-out sampling strategy, held-out set size, and compute overhead are each ablated separately, a discipline not common in this subfield.

Weaknesses:

1.	Dependence on labeled held-out data: The paper acknowledges this limitation but does not fully address its practical severity. In many real-world unlearning scenarios, access to labeled held-out data from the same distribution may not be available. While the authors suggest future work on external datasets or synthetic data, this is a significant deployment barrier. The held-out set also must be disjoint from the training data, which may not be guaranteed in practice.

2.	Limited theoretical grounding: The paper lacks formal connections between the distillation objective and the true goal of matching the retrained model. While empirical results are strong, there is no theoretical analysis of when/why the held-out reference distribution approximates the retrained model's behavior on forget samples. The relationship between the reference model (trained on the full dataset) and the desired retrained model is not characterized.

3.	Oracle teacher paradox: The ablation showing that the frozen base model performs similarly to the oracle retrained teacher raises questions about what exactly is being learned. If the frozen base teacher (which still "remembers" the forget data) yields comparable performance to the ideal retrained teacher, does this indicate that ReGUN is not truly removing the influence of forget data, but rather achieving MIA indistinguishability through other means? The membership inference results suggest effective forgetting, but the teacher ablation merits deeper discussion.

4.	MIA evaluation limitations: While RMIA is more sophisticated than simple loss-based attacks, the paper relies on a specific attack framework. Recent work has shown that more powerful MIAs can detect subtle membership signals even when standard attacks fail. The paper would benefit from evaluating against stronger attacks or discussing this limitation.

5.	Forget fraction sensitivity: The performance gap of ReGUN relative to baselines varies significantly across forget fractions. At 1% forgetting, many methods perform similarly, while at 50%, some specialized baselines (e.g., SALUN, AMUN) become more competitive. The paper should discuss when ReGUN is most advantageous and when simpler methods might suffice.

6.	Hyperparameter sensitivity: The paper uses per-method hyperparameter tuning but does not thoroughly analyze ReGUN's sensitivity to the retain/forget weight w and learning rate. Given that Figure 2 sweeps w for several methods, including this analysis for ReGUN across different settings would strengthen the paper.

**Audience:**

Yes

**Audience Explanation:**

This paper addresses a core tension in machine unlearning between forgetting efficacy and utility preservation, which is highly relevant to privacy and compliance in deployed AI systems.

**Claims And Evidence:**

No

**Claims Explanation:**

The core claim that the reference distribution approximates the oracle retrained model's behavior is not fully supported by the evidence, as the paper demonstrates that a frozen model (which still contains the forgotten data) performs equally well as the oracle, yet it never directly measures or verifies the similarity between these reference distributions to confirm they actually represent the same "unseen behavior."

**Requested Changes:**

1.	Include Bad Teacher and SCRUB as empirical baselines under the same protocol, or explain their exclusion: The methods the paper itself identifies as closest, Bad Teacher and SCRUB, are not empirically compared. A dedicated passage in Related Work distinguishes ReGUn from both conceptually, but this comparison remains textual; neither is included among the evaluated baselines, which instead skew toward degradation- and localization-based approaches. Without this empirical counterpart, the central claim of outperforming existing methods is incomplete.

2.	While ReGUN is an approximate method, do you foresee any possibility of extending it to provide formal unlearning guarantees (e.g., differential privacy-style certificates)? If not, what are the fundamental barriers?

3.	The abstract and conclusion's framing that a simple objective outperforms more complex methods is not fully consistent with the reported results: in at least one complete setting, a simpler existing method achieves a better aggregate score than ReGUn (Table 1, 10% forgetting, ResNet-18/CIFAR-10). This does not undermine the overall conclusion, but the narrative should more precisely reflect the data.

4.	Reference teacher choice: The frozen base model performs surprisingly well compared to the oracle retrained teacher. Could you elaborate on why this might be? Is it possible that the base model's predictions on held-out data already approximate the retrained model's behavior sufficiently well due to the held-out data being "unseen" by the base model?

5.	The reference-model ablation shows that a retrained oracle offers no advantage over the frozen initial model. While this suggests robustness to reference-model choice, the similarity between the resulting reference distributions is not directly measured, so the claim that the reference approximates oracle unseen behavior remains unverified.